# Efficient Interpolation between Extragradient and Proximal Methods for Weak MVIs

**Thomas Pethick**[*]      **Ioannis Mavrothalassitis**[*]      **Volkan Cevher**[*]

## Abstract

We study nonmonotone games satisfying the weak Minty variational inequality (MVI) with parameter $\rho \in (-1/L, \infty)$, where $L$ is the Lipschitz constant of the gradient operator. An error corrected version of the inexact proximal point algorithm is proposed, with which we establish the first $\mathcal{O}(1/\epsilon)$ rate for the entire range $\rho \in (-1/L, \infty)$, thus removing a logarithmic factor compared with the complexity of existing methods. The scheme automatically selects the needed accuracy for the proximal computation, and can recover the relaxed extragradient method when $\rho > -1/2L$ and the relaxed proximal point algorithm (rPPA) when $\rho > -1/L$. Due to the error correction, the scheme inherits the strong properties of the *exact* rPPA. Specifically, we show that linear convergence is automatically achieved under appropriate conditions. Tightness for the range of $\rho$ is established through a lower bound for rPPA. Central to the algorithmic construction is a halfspace projection, where the key insight is that the allowed error tolerance can both be used to correct for the proximal approximation and to enlarge the problem class.

## 1 Introduction

Nonconvex-nonconcave minimax problem—and more generally nonmonotone games—are ubiquitous in machine learning applications but notoriously difficult to solve (Daskalakis et al., 2021; Hirsch & Vavasis, 1987; Papadimitriou, 1994; Hsieh et al., 2021). A nonmonotone class that has attracted significant attention is the class of problems satisfying the weak Minty variational inequality (MVI) introduced in Diakonikolas et al. (2021). One reason for the increased interest is that the counterexamples constructed to demonstrate the failure of classical methods in Hsieh et al. (2021) were shown to satisfy the weak MVI (Pethick et al., 2022).

The weak MVI is dictated by a parameter $\rho$, which determines the degree of allowed nonmonotonicity. A flurry of work has followed since Diakonikolas et al. (2021) focusing on extending the range of $\rho$ (Pethick et al., 2022; Lee & Kim, 2024; Fan et al., 2023; Alacaoglu et al., 2024; Pethick et al., 2024). At the heart of all these methods is a *relaxed* update rule. Interestingly, the relaxed extragradient method achieves a (tight) $\rho \in (-1/2L, \infty)$ (Pethick et al., 2022; 2024) while the relaxed proximal point algorithm (rPPA) permits the larger range of $\rho \in (-1/L, \infty)$ (Alacaoglu et al., 2024; Pethick et al., 2024). In other words, moving beyond $\rho > -1/2L$ seems to require an increasingly refined approximation of the proximal operator made precise by the logarithmic factor suffered in the complexity (Alacaoglu et al., 2024). This is in stark contrast with monotone problems where the extragradient method (Korpelevich, 1977) can approximate the proximal point algorithm without any tradeoff (Solodov & Svaiter, 1999b; Mokhtari et al., 2020).

The paper addresses the following fundamental question:

*Can we treat the weak MVI range $\rho \in (-1/L, -1/2L]$ without suffering a logarithmic factor?*   (⋆)

Very recently Fan et al. (2023) interestingly managed to show convergence for $\rho > -\frac{1 - 1/e}{L} \approx -\frac{0.632}{L}$ in unconstrained problems without a logarithmic factor in the complexity. However, it is unclear if the approach can be generalized to the larger range of $\rho$ and constrained settings.

---

[*]Laboratory for Information and Inference Systems (LIONS), EPFL (thomas.pethick@epfl.ch)

Table 1: Comparison with existing literature. Algorithm 1 is the first method to treat weak MVI with $\rho > -1/L$ without suffering a logarithmic factor in the complexity.

| Method | Minimum $\rho$ | Complexity[1] | Constraints | Linear convergence[2] |
|---|---|---|---|---|
| Diakonikolas et al. (2021) | $-\frac{1}{8L}$ | $\mathcal{O}(\frac{1}{\varepsilon})$ | ✗ | ✗ |
| Pethick et al. (2022) | $-\frac{1}{2L}$ | $\mathcal{O}(\frac{1}{\varepsilon})$ | ✓ | ✗ |
| Böhm (2022) | $-\frac{1}{2L}$ | $\mathcal{O}(\frac{1}{\varepsilon})$ | ✗ | ✗ |
| Fan et al. (2023) | $-\frac{1-1/e}{L}$ | $\mathcal{O}(\frac{1}{\varepsilon})$ | ✗ | ✗ |
| Alacaoglu et al. (2024) | $-\frac{1}{L}$ | $\mathcal{O}(\frac{1}{\varepsilon}\ln\frac{1}{\varepsilon})$ | ✓ | ✗ |
| **This paper** | $-\frac{1}{L}$ | $\mathcal{O}(\frac{1}{\varepsilon})$ | ✓ | ✓ |

[1]Number of operator evaluations for making the squared tangent residual smaller than $\varepsilon$.
[2]Linear convergence is established under an error bound condition (cf. Section 7).

We answer (⋆) in the affirmative. Our analysis generalizes the halfspace projection approaches in both Pethick et al. (2022) and Solodov & Svaiter (1999a). The key observation is that the allowed error tolerance in the halfspace projection can both be used to expand the problem class and to correct for the inexactness of the proximal operator estimation. Our construction provides an intuitive geometric explanation for convergence results in weak MVIs.

Concretely we make the following contributions:

(i) **Improved complexity** We propose a hybrid proximal extragradient method (Algorithm 1) to achieve convergence for $\rho > -1/L$, which automatically selects the required approximation quality for the proximal operator through a computationally negligible error condition. The stopping criterion combined with the error correction step in Algorithm 1 removes the logarithmic factor in the complexity, while generalizing to constrained and regularized settings. The tightness of the range of $\rho$ is made precise through a lower bound for the idealized case of *exact* proximal computations.

(ii) **Unification** The error condition for the inner loop in Algorithm 1 can pass immediately if $\rho > -1/2L$, in which case the scheme exactly reduces to the AdaptiveEG+ method of Pethick et al. (2022). Algorithm 1 is an instantiation of the implicit scheme (8) which can recover the hybrid method of Solodov & Svaiter (1999a) and the celebrated forward-backward-forward algorithm (Tseng, 2000) when the problem is monotone (i.e. $\rho = 0$). The scheme thus unifies the analysis of the relaxed extragradient method, the relaxed proximal point algorithm (rPPA) and classical methods for monotone problems, while providing a precise explanation for why the relaxed extragradient method only applies to $\rho > -1/2L$.

(iii) **Linear convergence** Algorithm 1 adopts strong properties of the exact rPPA due to the error correction. Specifically Algorithm 1 automatically obtains linear convergence under appropriate conditions by exploiting the Fejér monotonicity of the iterates. Additionally, Fejér monotonicity simplifies the proofs in general by saving us from meticulously arguing about boundedness of the iterates otherwise necessary in e.g. Alacaoglu et al. (2024); Pethick et al. (2024). The simplicity is apparent as the proof for the general case fits on half a page, even in the adaptive case (cf. Appendix C) and in 5 lines in the nonadaptive case (cf. Theorem 5.5).

## 2 RELATED WORK

Variational inequalities (VIs) provide a unifying framework for studying optimization, equilibrium, and fixed point problems (Facchinei & Pang, 2003). Classical work has focused on monotone VIs, where the operator satisfies a global monotonicity condition. For such problems, a wide range of efficient algorithms have been developed, including the (inexact) proximal point method (Rockafellar, 1976), the extragradient method (Korpelevich, 1977), and the forward-backward-forward method (Tseng, 2000).

Recent works have begun to study nonmonotone VIs motivated by nonconvex-nonconcave minimax optimization. Diakonikolas et al. (2021) introduced the weak Minty variational inequality (MVI)

and showed convergence of the relaxed extragradient method for $\rho > -1/8L$. Subsequent works have focused on pushing the allowable range of $\rho$ using various algorithmic techniques. Pethick et al. (2022) extended the definition to constrained problems and proposed an algorithm with convergence for the range $\rho > -1/2L$. Fan et al. (2023) proposed an extension using multiple forward operator evaluation, $H := \mathrm{id} - \gamma F$, per iteration, extending the range to $\rho > -1-1/e/L$ for unconstrained problems. Lee & Kim (2024) also extended the range of $\rho$ using a hyperplane projection, but at the cost of a logarithmic factor in the complexity. It turns out that relaxed (inexact) proximal point methods (Pethick et al., 2024; Alacaoglu et al., 2024) can achieve the relaxed range $\rho > -1/L$ as first shown in Alacaoglu et al. (2024). The same range can be obtained by applying the inexact proximal point method in e.g. Chen & Luo (2022) to the modified operator defined in Lee & Kim (2021, App. A.1). Adaptive approaches has also been taken in an attempt to increase the range of $\rho$ even further (Pethick et al., 2022; Böhm, 2022; Alacaoglu et al., 2023).

Whereas the previous schemes can be seen as instances of the Krasnosel'skiĭ-Mann (KM) iteration from the fixed point literature (Pethick et al., 2024), there exists another line of work focusing on the Halpern iteration in the special case of cohypomonotone problems. In contrast with the KM iteration, Halpern iteration linearly interpolates with the initial point using a time-varying stepsize, i.e. $z^{k+1} = (1-\lambda_k)z^0 - \lambda_k T z^k$. This construction has been used to show optimal $\mathcal{O}(1/k^2)$ convergence rates for the squared fixed point residual. Sabach & Shtern (2017); Lieder (2021) originally showed convergence for nonexpansive operators, which implies convergence for $\rho$-comonotone problems with $\rho > -1/2L$ using the (exact) proximal operator. By directly approximating the proximal operator, an explicit scheme was later proposed for monotone problems in Diakonikolas (2020), suffering a logarithmic factor in the rate. The logarithmic factor was removed for unconstrained problems by means of an extragradient variant in Yoon & Ryu (2021), which was later extended to $\rho$-cohypomonotone problems with $\rho > -1/2L$ in Lee & Kim (2021) and subsequently the constrained case in Cai et al. (2022a) while only requiring a single projection. Recently, Alacaoglu et al. (2024) showed convergence for the range $\rho > -1/L$ using the inexact Halpern iteration by exploiting the weaker structure of conic-nonexpansiveness (Bauschke et al., 2021; Bartz et al., 2022).

## 3    PRELIMINARIES

We will formulate our problem as a (possibly nonmonotone) inclusion problem in which we seek to find $z \in \mathbb{R}^d$ such that

$$0 \in Sz := Fz + Az. \tag{1}$$

More compactly, we will write $z \in \mathrm{zer}\, S$. We will use the single-valued operator $F : \mathbb{R}^d \to \mathbb{R}^d$ to capture the smooth part of the problem and $A : \mathbb{R}^d \rightrightarrows \mathbb{R}^d$ to capture (projectable) constraints as made precise in the following assumptions (cf. Appendix A for missing definitions).

**Assumption 3.1.** *In problem* (1),

(i) *The operator* $F : \mathbb{R}^d \to \mathbb{R}^d$ *is* $L$-*Lipschitz, i.e. for some* $L \in [0, \infty)$,

$$\|Fz - Fz'\| \leq L\|z - z'\| \quad \forall z, z' \in \mathbb{R}^d. \tag{2}$$

(ii) *The operator* $A : \mathbb{R}^d \rightrightarrows \mathbb{R}^d$ *is maximally monotone.*

(iii) *The operator* $S = F + A$ *satisfies the weak Minty variational inequality (MVI), i.e. there exists a nonempty solution set* $\mathcal{Z}^\star \subseteq \mathrm{zer}\, S$ *such that for all* $z^\star \in \mathcal{Z}^\star$ *and some* $\rho \in \left(-\frac{1}{L}, \infty\right)$

$$\langle v, z - z^\star \rangle \geq \rho\|v\|^2 \quad \forall(z, v) \in \mathrm{gph}\, S. \tag{3}$$

One prominent problem class that can be cast as the inclusion (1) is $m$-player games.

**Example 3.2** ($m$-player games). *Denote the decision variables* $z := (z_i; z_{-i}) := (z_1, ..., z_m) \in \mathbb{R}^d$ *with* $d = \sum_{i=1}^m d_i$ *and let the loss incurred by the* $i^{\mathrm{th}}$ *player be* $\mathcal{L}_i(z_i; z_{-i}) = \varphi_i(z) + g_i(z_i)$ *where* $\varphi_i$ *is the payoff function and* $g_i$ *typically enforce constraints on* $z_i$. *A Nash equilibrium is any decision* $z^\star \in \mathbb{R}^d$ *which is unilaterally stable, i.e.,*

$$\mathcal{L}_i(z_i^\star; z_{-i}^\star) \leq \mathcal{L}_i(z_i; z_{-i}^\star) \quad \forall z_i \in \mathbb{R}^{d_i} \text{ and } i \in [m] := \{1, \ldots, m\}. \tag{4}$$

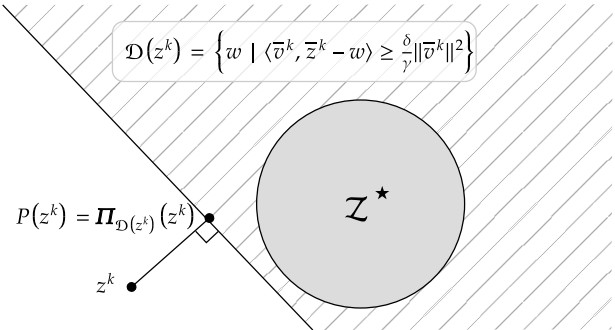

Figure 1: The scheme (8) proceeds by iteratively projecting onto the constructed halfspace $\mathcal{D}(z^k)$, which is guaranteed to contain the solution set $\mathcal{Z}^\star \subseteq \text{zer } S$. It turns out that only a *relative* error condition when computing $\bar{z}^k$ is sufficient to ensure that the sequence do not terminate prematurely.

*The corresponding first order optimality conditions may be written as the inclusion* (1) *with* $Fz = (\nabla_{z_1}\varphi_1(z), \ldots, \nabla_{z_m}\varphi_m(z))$ *and* $Az = (\partial g_1(z_1), \ldots, \partial g_m(z_m))$. *The operator $A$ will only be accessed through the resolvent*

$$J_{\gamma A}(z) := (\text{id} + \gamma A)^{-1}(z) = (\text{prox}_{\gamma g_1}(z_1), \ldots, \text{prox}_{\gamma g_m}(z_m))$$

*with $\gamma > 0$. Maximal monotonicity of $A$ is satisfied when $g_i$ is proper lsc convex for all $i \in [m]$.*

## 4 ALGORITHMIC CONSTRUCTION

Our starting point is the (inexact) proximal update on $S := F + A$, which given a $z \in \mathbb{R}^d$ seeks:

$$\bar{z} \in (\text{id} + \gamma S)^{-1}(z - \varepsilon) \quad \Leftrightarrow \quad \bar{z} = z - (\bar{v} + \varepsilon) \quad \text{and} \quad \bar{v} \in \gamma S\bar{z} \tag{5}$$

with inexactness $\varepsilon \in \mathbb{R}^d$. The above update can be applied iteratively, $z^{k+1} = (\text{id} + \gamma S)^{-1}(z^k - \varepsilon^k)$ when the resolvent is singled-valued, as classically considered (Rockafellar, 1976), but establishing convergence would require the errors to be absolutely summable, i.e. $\sum_{k=0}^{\infty} \|\varepsilon^k\| < \infty$, which translates into an increasing number of inner iterations used to approximate the proximal operator.

Instead we will introduce a halfspace projection to correct for the inexactness in the proximal evaluation, which we will refer to interchangeably as the *error correction* step. The construction is largely inspired by Solodov & Svaiter (1999a), which introduced the idea for monotone problems to relax the standard absolute summable error criterion to only requiring what was coined as *relative* inexactness. In what follows, we will importantly construct a halfspace that can be used for the more general class of weak MVI.

Observe that, regardless of the inexactness, the proximal update (5) can be used to construct a halfspace that contains the solution set in Assumption 3.1(iii)

$$\mathcal{D}(z) = \{\, w \mid \langle \bar{v}, \bar{z} - w \rangle \geq \tfrac{\delta}{\gamma} \|\bar{v}\|^2 \,\} \tag{6}$$

where $\delta \leq \rho$. The containment of the solution set is easily verified by taking $w \in \mathcal{Z}^\star$. If we can iteratively project onto this (changing) halfspace, then convergence immediately follows, provided that we can argue that there is no fixed point other than $z \in \text{zer } S$. For later convenience we define the projection operator $P$. As we will see, it is fairly easy to derive the closed form solution to this halfspace projection, which effectively results in using an extragradient computation $\bar{v} \in \gamma S\bar{z}$:

$$P(z) := \mathbf{\Pi}_{\mathcal{D}(z)}(z) = \underset{u \in \mathcal{D}(z)}{\arg\min} \|u - z\| = \begin{cases} z - \alpha\bar{v} & z \notin \mathcal{D}(z) \\ z & z \in \mathcal{D}(z) \end{cases} \quad \text{with} \quad \alpha = \frac{\langle \bar{v}, z - \bar{z} \rangle + \frac{\delta}{\gamma}\|\bar{v}\|^2}{\|\bar{v}\|^2}. \tag{7}$$

A convergent sequence can readily be constructed by repeatedly applying the projection, i.e. $z^{k+1} = P(z^k)$. This iterative scheme is known as the Picard iterations in the fixed point iteration literature, and it converges for firmly (quasi)-nonexpansive operators—a property which holds for the projection operator $P$ as formally established in Proposition 5.1.

---

**Algorithm 1** An explicit hybrid proximal extragradient method

---

**Require:** $z^0 \in \mathbb{R}^d$ $\lambda_k \in (0,2), \sigma \in [0, 1 + \frac{\delta}{\gamma}), \delta \leq \rho, \gamma \in (\lfloor -\rho \rfloor_+, 1/L)$
**Repeat** for $k = 0, 1, \ldots$ until convergence
1: $\bar{z}^k \leftarrow z^k$
2: **repeat**
3:     $h^k \leftarrow z^k - \gamma F\bar{z}^k$
4:     $\bar{z}^k \leftarrow (\mathrm{id} + \gamma A)^{-1} h^k$
5: **until** $\langle z^k - \bar{z}^k, \bar{v}^k \rangle \geq (1 - \sigma)\|\bar{v}^k\|^2$ where $\bar{v}^k = h^k - \bar{z}^k + \gamma F\bar{z}^k$
6: $z^{k+1} = z^k - \lambda_k \alpha_k \bar{v}^k$ with $\alpha_k = \frac{\langle \bar{v}^k, z^k - \bar{z}^k \rangle}{\|\bar{v}^k\|^2} + \frac{\delta}{\gamma}$

**Return** $z^{k+1}$

---

In order to later relate our adaptive scheme to existing fixed variants in the literature, it will prove useful to introduce an (over)relaxation parameter $\lambda_k \in (0,2)$. The resulting scheme is known as the Krasnosel'skiĭ-Mann (KM) iteration, which recovers the Picard iteration for $\lambda_k = 1$:

$$z^{k+1} = (1 - \lambda_k)z^k + \lambda_k P(z^k) \tag{KM}$$

with $\lambda_k > 0$. It is worth highlighting that our use of the above relaxation is different from the one used in Alacaoglu et al. (2024); Pethick et al. (2024). They apply the KM iteration directly to the (inexact) proximal operator, and rely on the (under)relaxation to establish convergence for weak MVIs with $\rho > -\frac{1}{L}$. We instead apply it to the halfspace projection operator $P$ and only use the generalization to expand the algorithmic class. In particular, we will still be able to show convergence for $\rho > -\frac{1}{L}$ even when $\lambda_k \in [1, 2)$.

By expanding the operator $P$ in KM and including the sufficient error condition (12) from Proposition 5.1 we obtain the following algorithm with $\gamma > 0$, $\lambda_k \in (0, 2)$, $\delta \leq \rho$ and $\sigma \in [0, 1 + \frac{\delta}{\gamma})$:

$$
\begin{aligned}
\text{find} \quad & \bar{z}^k \in \mathbb{R}^d \quad \text{and} \quad \bar{v}^k \in \gamma S\bar{z}^k \\
\text{s.t.} \quad & \bar{z}^k = z^k - (\bar{v}^k + \varepsilon^k) \quad \text{and} \quad -\langle \varepsilon^k, \bar{v}^k \rangle \leq \sigma\|\bar{v}^k\|^2 \\
\text{update} \quad & z^{k+1} = z^k - \lambda_k \alpha_k \bar{v}^k \quad \alpha_k = \frac{\langle \bar{v}^k, z^k - \bar{z}^k \rangle}{\|\bar{v}^k\|^2} + \frac{\delta}{\gamma}
\end{aligned}
\tag{8}
$$

*Remark* 4.1. The scheme generalizes both Pethick et al. (2022, Alg. 1) and the construction in Solodov & Svaiter (1999a). To recover Pethick et al. (2022, Alg. 1) take $\varepsilon^k = \gamma(Fz^k - F\bar{z}^k)$ for which the error condition can automatically pass for $\rho > -\gamma/2$ (see Section 6). To recover Solodov & Svaiter (1999a), which applies to the monotone case ($\rho = 0$), we first choose $\delta = 0$. Furthermore, from the Cauchy-Schwarz inequality and the update rule we recover the more stringent error condition $\|\varepsilon^k\| \leq \sigma \max\{\|\bar{v}^k\|, \|z^k - \bar{z}^k\|\}$ from Solodov & Svaiter (1999a). For $\varepsilon^k = 0$ we obtain the relaxed proximal point algorithm (Eckstein & Bertsekas, 1992) with the relaxation parameter $\bar{\alpha}_k = \lambda_k \alpha_k = \lambda_k(1 + \frac{\delta}{\gamma})$. See Appendix B for an overview of the special cases.

**Comparison** Alacaoglu et al. (2024); Pethick et al. (2024) also considers an inexact relaxed proximal point update. For any other choice than $\varepsilon^k = 0$, the algorithm (8) differs from the approach in Alacaoglu et al. (2024); Pethick et al. (2024) by not using $\bar{z}^k$ directly in the update of $z^{k+1}$ but rather importantly the *extragradient* evaluation $\bar{v}^k \in \gamma S\bar{z}^k$. This will turn out to be crucial to establishing Fejér monotonicity and thus avoiding the logarithmic factor in the complexity. Specifically, the update in Alacaoglu et al. (2024); Pethick et al. (2024) is the relaxed inexact proximal point algorithm

$$
\begin{aligned}
\bar{z}^k &\simeq (\mathrm{id} + \gamma S)^{-1}(z^k) \\
z^{k+1} &= (1 - \lambda_k)z^k + \lambda_k \bar{z}^k
\end{aligned}
\tag{9}
$$

where $\lambda_k \in (0, 1)$. In contrast, our proposed (8) uses an extragradient evaluation at $\bar{z}^k$ as follows:

$$
\begin{aligned}
\bar{z}^k &\simeq (\mathrm{id} + \gamma S)^{-1}(z^k) \\
z^{k+1} &= (1 - \lambda_k)z^k + \lambda_k(z^k - \alpha_k \bar{v}^k)
\end{aligned}
\tag{10}
$$

where the extragradient evaluation $\bar{v}^k \in \gamma S\bar{z}^k$ is defined through (8). We further discuss the difference with existing schemes in Section 6.3.

**Explicit scheme**  To derive an explicit scheme we will define an inner iteration to approximating the resolvent $(\mathrm{id} + \gamma S)^{-1}$. Recall $S := A + F$. Given $z \in \mathbb{R}^d$ we seek $z' \in \mathbb{R}^d$ such that

$$z' = (\mathrm{id} + \gamma S)^{-1}z = (\mathrm{id} + \gamma A)^{-1}(z - \gamma F z')$$

Following Nemirovski (2004); Pethick et al. (2024) this can be approximated with a fixed point iteration of

$$C_z : w \mapsto (\mathrm{id} + \gamma A)^{-1}(z - \gamma F w) \tag{11}$$

which is a contraction for small enough $\gamma$ since $F$ is Lipschitz continuous. It follows from Banach's fixed-point theorem Banach (1922) that the sequence converges linearly. Approximating $\bar{z}^k$ in (8) with an inner iteration (11) leads to Algorithm 1.

## 5   ANALYSIS OF THE IMPLICIT SCHEME (8)

This section establishes convergence of the implicit scheme (8). The key to showing convergence is to guarantee that we always make progress towards a zero of $S$ when applying the halfspace projection operator $P$ in (7). Specifically we will argue that the numerator $\langle \bar{v}, z - \bar{z}\rangle + \frac{\delta}{\gamma}\|\bar{v}\|^2$ in the (adaptive) stepsize $\alpha$ remains positive even with the inaccuracy $\varepsilon$ (such that fix $P \subseteq \mathcal{Z}^\star$, which prevents the sequence from terminating prematurely). We now show crucial properties of this operator.

**Proposition 5.1** (Properties of (7))**.** *Suppose Assumption 3.1(iii) holds, $\delta \leq \rho$, and (5) satisfies the following error condition,*

$$-\langle \varepsilon, \bar{v}\rangle \leq \sigma\|\bar{v}\|^2. \tag{12}$$

*where $\sigma \in [0, 1 + \frac{\delta}{\gamma})$. Then,*

*(i) The projection operator $P : \mathbb{R}^d \to \mathbb{R}^d$ in (7) is firmly quasi-nonexpansive.*

*(ii) $\mathcal{Z}^\star \subseteq \mathrm{fix}\,P \subseteq \mathrm{zer}\,S$.*

*(iii) The closed form solution to $P$ is given as in (7) and the stepsize satisfies $\alpha \geq 1 + \frac{\delta}{\gamma} - \sigma$.*

*Remark* 5.2.  Notice how the approximation quality in (12) needs to be better (while still only relative) for increasingly negative $\rho$ through the requirement $\sigma < 1 + \frac{\delta}{\gamma} \leq 1 + \frac{\rho}{\gamma}$. Trivially, when $\sigma = 0$ we get the loose requirement of $\rho > -\gamma > -\frac{1}{L}$ where the last inequality follows from the requirement $\gamma < 1/L$ for the proximal operator to be single-valued (see Lemma C.2 and Remark 5.4).

Convergence of (8) follows from convergence of the KM iteration (see Theorem C.1) applied to the firmly quasi-nonexpansive operator $P$ and the fact that fix $P \subseteq \mathrm{zer}S$. The following theorem is a direct consequence of the previous Proposition 5.1 and Theorem C.1. The theorem establishes a convergence rate for what is referred to as the tangent residual (Cai et al., 2022b;a; Bot et al., 2023).

**Theorem 5.3.** *Suppose Assumption 3.1(iii) holds. Consider the sequence $(z^k)_{k\in\mathbb{N}}$ generated by (8) with $\lambda_k \in (0, 2)$, $\kappa := \liminf_{k\to\infty} \lambda_k(2 - \lambda_k) > 0$, $\delta \leq \rho$, and $\sigma \in [0, 1 + \frac{\delta}{\gamma})$. Then, for all $z^\star \in \mathcal{Z}^\star$*

$$\min_{k\in\{0,\dots,K-1\}} \mathrm{dist}(0, S\bar{z}^k)^2 \leq \frac{\|z^0 - z^\star\|^2}{\kappa\gamma^2(1 + \frac{\delta}{\gamma} - \sigma)^2 K}.$$

*Furthermore, if $\mathcal{Z}^\star = \mathrm{zer}\,S$ then $(z^k)_{k\in\mathbb{N}}$ converges to some $z^\star \in \mathcal{Z}^\star$.*

*Remark* 5.4.  It is important to ensure that the (inexact) proximal update is well-defined. The exact case holds under maximal monotonicity of the operator $A$ and Lipschitz continuity of the operator $F$ when $\gamma < \frac{1}{L}$ (see Lemma C.2). We have assumed maximal monotonicity of $A$ for simplicity. However, note that it is trivially possible to relax the condition on $A$ to cohypomonotonicity when $S := F + A$ is also cohypomonotonicity (with $\rho > -\gamma$) since well-definedness of $J_{\gamma S}$ is preserved (Bauschke et al., 2021, Cor. 2.14). The inexact case will be treated for the particular choice of inexactness, since we will importantly be able to relax the stepsize requirement to $\gamma \leq \frac{1}{L}$.

Convergence for a variant of (8) with a nonadaptive relaxation parameter $\lambda_k\alpha_k = \bar{\alpha}$ follows immediately, since the adaptive stepsize can be absorbed into $\lambda_k \in (0, 2)$ as remarked in Section 6.3. However, we include a direct proof as it turns out to be particularly compact.

**Theorem 5.5.** *Suppose Assumption 3.1(iii) holds. Consider the sequence $(z^k)_{k \in \mathbb{N}}$ generated by (8) with the nonadaptive stepsize $\lambda_k \alpha_k = \bar{\alpha} \in (0, 2(1 - \sigma + \frac{\rho}{\gamma}))$. Then, for all $z^\star \in \mathcal{Z}^\star$*

$$\min_{k \in \{0, \ldots, K-1\}} \mathrm{dist}(0, S\bar{z}^k)^2 \leq \frac{\|z^0 - z^\star\|^2}{2\gamma^2 \bar{\alpha}(1 - \sigma - \frac{\bar{\alpha}}{2} + \frac{\rho}{\gamma})K}.$$

*Proof.* We proceed by expanding the update for rule in (8) and establishing Fejér monotonicity:

$$
\begin{aligned}
\|z^{k+1} - z^\star\|^2 &= \|z^k - z^\star\|^2 + \bar{\alpha}^2 \|\bar{v}^k\|^2 - 2\bar{\alpha}\langle \bar{v}^k, z^k - z^\star \rangle \\
&= \|z^k - z^\star\|^2 + \bar{\alpha}^2 \|\bar{v}^k\|^2 - 2\bar{\alpha}\langle \bar{v}^k, z^k - \bar{z}^k \rangle - 2\bar{\alpha}\langle \bar{v}^k, \bar{z}^k - z^\star \rangle \\
&= \|z^k - z^\star\|^2 - 2\bar{\alpha}(1 - \tfrac{\bar{\alpha}}{2})\|\bar{v}^k\|^2 - 2\bar{\alpha}\langle \bar{v}^k, \varepsilon^k \rangle - 2\bar{\alpha}\langle \bar{v}^k, \bar{z}^k - z^\star \rangle \\
\text{(Assumption 3.1(iii))} \quad &\leq \|z^k - z^\star\|^2 - 2\bar{\alpha}(1 - \tfrac{\bar{\alpha}}{2} + \tfrac{\rho}{\gamma})\|\bar{v}^k\|^2 - 2\bar{\alpha}\langle \bar{v}^k, \varepsilon^k \rangle \\
\text{(error condition)} \quad &\leq \|z^k - z^\star\|^2 - 2\bar{\alpha}(1 - \sigma - \tfrac{\bar{\alpha}}{2} + \tfrac{\rho}{\gamma})\|\bar{v}^k\|^2 \qquad (13)
\end{aligned}
$$

Rearranging, summing and telescoping completes the proof. $\qquad\square$

A few remarks are in place. First notice that there is almost no looseness in the proof as we only apply an inequality once, which is the weak MVI that we work under—the error condition does not add any looseness as we can control its tightness. Secondly, from (13), the interplay between the relaxation parameter $\bar{\alpha}$, the error tolorance $\sigma$ and the nonmonotonicity constant $\rho$ becomes very apparent. Specifically, the parameter $\bar{\alpha}$ needs to be taken smaller with increasingly negative $\rho$ and large error tolerance $\sigma$. Note that Theorem 5.5 also implies convergence of the explicit Algorithm 1 when we force $\lambda_k \alpha_k = \bar{\alpha}$.

## 6 ANALYSIS OF THE EXPLICIT SCHEME (ALGORITHM 1)

Algorithm 1 is an instance of (8) with a particular iterative approximation of $\bar{z}^k$. Thus, in order to obtain convergence it suffice to show that the error condition will eventually pass. Due to Pethick et al. (2024, Lem. 5.1), which analyses the same inner iteration as Algorithm 1, the error $\|\varepsilon^k\|$ can be made arbitrarily small, which would be sufficient for ensuring the error condition.

In this section we will show a stronger result by characterizing the number of inner iterations in Algorithm 1 sufficient for convergence of a given $\rho$, and crucially show that the number of inner iterations can be $\mathcal{O}(1)$ instead of only $\mathcal{O}(\ln \frac{1}{\epsilon})$. As a warmup we first show that a single inner iterations suffice when $\rho > -\frac{1}{2L}$, thus exactly recovering the result of Pethick et al. (2022).

### 6.1 SINGLE-STEP APPROXIMATION

We will show that a single-step approximation of the resolvent suffice under the more stringent requirement of $\rho > -\frac{\gamma}{2} > -\frac{1}{2L}$. To obtain the direct extragradient type scheme in Pethick et al. (2022, Alg. 1) take the error in (8) to be $\varepsilon^k = \gamma(Fz^k - F\bar{z}^k)$ such that the update rule defining the extrapolated point $\bar{z}^k$ can be explicitly computed with one forward-backward step

$$\bar{v}^k = Hz^k - H\bar{z}^k \quad \Leftrightarrow \quad \bar{z}^k = (\mathrm{id} + \gamma A)^{-1} Hz^k \qquad (14)$$

where $H := \mathrm{id} - \gamma F$ is the forward operator and well-definedness of the resolvent follows from maximally monotonicity of $A$. In turn, the error condition in (8) reduces to

$$-\gamma \langle Fz^k - F\bar{z}^k, Hz^k - H\bar{z}^k \rangle \leq \sigma \|Hz^k - H\bar{z}^k\|^2.$$

We want to understand for what $\sigma$ this condition is met. From $\frac{1}{2}$-cocoercivity of $H$ (Lemma A.4(i)) we have the following when the operator $F : \mathbb{R}^d \to \mathbb{R}^d$ is Lipschitz continuous and $\gamma \leq \frac{1}{L}$

$$\tfrac{1}{2}\|Hz^k - H\bar{z}^k\|^2 \leq \langle Hz^k - H\bar{z}^k, z^k - \bar{z}^k \rangle = \gamma \langle Hz^k - H\bar{z}^k, Fz^k - F\bar{z}^k \rangle + \|Hz^k - H\bar{z}^k\|^2.$$

Consequently, the error condition holds with $\sigma \in [\frac{1}{2}, 1 + \frac{\rho}{\gamma})$ in which case $\rho > -\frac{\gamma}{2}$ is sufficient for convergence.

Notice that the update (14) correspond to having one inner step of Algorithm 1. Thus, by the same argument the error condition in Algorithm 1 will immediately pass if $\sigma \in [\frac{1}{2}, 1 + \frac{\rho}{\gamma})$ and Algorithm 1 exactly reduces to Pethick et al. (2022, Alg. 1). The statement is made precise in Theorem 6.1(i).

## 6.2 MULTIPLE INNER STEPS

The key to extending the nonmonotonicity parameter $\rho$ is observing that we can push the *lower bound* on the relative inexactness parameter $\sigma \in (0, 1 + \frac{\rho}{\gamma})$, for which the error condition is guaranteed to pass, closer to zero by increasing the number of inner iterations. Consequently, the condition on $\rho$ becomes weaker. The following theorem makes this precise, where the main technical difficulty comes from having to establish a lower bound on $\|\bar{v}^k\|^2$ appearing in the error condition. The convergence rate in the theorem is a direct consequence of Theorem 5.3.

**Theorem 6.1.** *Suppose Assumption 3.1 holds. Consider the sequence $(z^k)_{k \in \mathbb{N}}$ generated by Algorithm 1 and let $\kappa := \liminf_{k \to \infty} \lambda_k(2 - \lambda_k) > 0$. Then, for all $z^\star \in \mathcal{Z}^\star$*

$$\min_{k \in \{0,\dots,K-1\}} \text{dist}(0, S\bar{z}^k)^2 \leq \frac{\|z^0 - z^\star\|^2}{\kappa \gamma^2 (1 + \frac{\delta}{\gamma} - \sigma)^2 K}.$$

*Furthermore, if $\mathcal{Z}^\star = \text{zer } S$ then $(z^k)_{k \in \mathbb{N}}$ converges to some $z^\star \in \mathcal{Z}^\star$. Moreover, let $n$ denote the number of inner iterations in Algorithm 1. Then,*

  (i) *the error condition passes immediately (i.e. $n = 1$) even when only $\gamma \leq \frac{1}{L}$ if $\sigma \in (\frac{1}{2}, 1 + \frac{\rho}{\gamma})$ which is feasible for $\rho > -\frac{\gamma}{2}$.*

  (ii) *the error condition passes after at most $n$ iterations for some $\gamma < 1/L$ if $\sigma \in (\frac{2 - \ln(n)/n}{n+1 - \ln(n)/n}, 1 + \frac{\rho}{\gamma})$ which is feasible for $\rho > -\frac{1}{L}(1 - \frac{\ln(n)}{n})\frac{n-1}{n+1 - \ln(n)/n}$.*

*Remark* 6.2. For simplicity of exposition we assume maximally monotonicity of $A : \mathbb{R}^d \rightrightarrows \mathbb{R}^d$, but the assumption can be relaxed to $\rho_A$-comonotonicity with $\rho_A > -\gamma/2$ without modifying the argument, since the resolvent $J_{\gamma A}$ remains nonexpansive (Bauschke et al., 2021, Prop. 3.13(iii)).

We can immediately infer complexity bounds from Theorem 6.1 without a logarithmic factor, due to the fact that the number of inner steps $n$ is $\mathcal{O}(1)$. The characterization interestingly uncovers an additional dependency on $\rho$ in the complexity in the interval $\rho \in (-\frac{1}{L}, -\frac{1}{2L}]$, which we conjecture is unavoidable considering the tightness of our construction.

**Corollary 6.3.** *Suppose Assumption 3.1 holds. Consider the sequence $(z^k)_{k \in \mathbb{N}}$ generated by Algorithm 1 and let $\kappa := \liminf_{k \to \infty} \lambda_k(2 - \lambda_k) > 0$. Then, for all $z^\star \in \mathcal{Z}^\star$ the sequence achieves $\min_{k \in \{0,\dots,K-1\}} \text{dist}(0, S\bar{z}^k)^2 \leq \epsilon$ after at most*

$$\#(oracle\ calls) \leq \frac{(1+n)\|z^0 - z^\star\|^2}{\kappa \gamma^2 (1 + \frac{\delta}{\gamma} - \sigma)^2 \epsilon}$$

*to both the operator $F$ and the resolvent $(\text{id} + \gamma A)^{-1}$ where $n = \lceil \frac{6}{1 + \rho L} \ln(\frac{3}{1 + \rho L}) \rceil$ is an upper bound on the number of inner iterations.*

*Remark* 6.4. Corollary 6.3 removes the logarithmic factor in $\epsilon$ that otherwise appears for approaches approximating the proximal operator (Pethick et al., 2024; Alacaoglu et al., 2024; Lee & Kim, 2024). Not only does Algorithm 1 improve the complexity from $\mathcal{O}(\frac{1}{\epsilon} \ln \frac{1}{\epsilon})$ to $\mathcal{O}(\frac{1}{\epsilon})$ for $\rho > -(1 - \sigma)\gamma$, but the method removes the need for hardcoding a potentially pessimistic choice of the number of inner steps and the stepsize $\alpha_k$ by instead chosen them based on the local structure of each iteration.

## 6.3 NONADAPTIVE VARIANT

Although the adaptive stepsize $\alpha_k$ and error condition of Algorithm 1 comes at essentially no additional computational cost, it will prove instructive to derive a scheme with a fixed number of inner iterations and a nonadaptive stepsize. Specifically, the variant will illuminate differences and similarities with the relaxed approximate proximal point method (RAPP) of Pethick et al. (2024) which can similarly converge for $\rho > -1/L$. The method is defined by the following recursion

$$\begin{aligned}
\bar{z}^k_{i+1} &= (\text{id} + \gamma A)^{-1}(z^k - \gamma F\bar{z}^k_i) \quad \forall i = 0, \dots, n-1 \quad \text{with} \quad \bar{z}^k_0 = z^k \\
z^{k+1} &= (1 - \lambda_k)z^k + \lambda_k \bar{z}^k_n
\end{aligned} \tag{RAPP}$$

with $\lambda_k \in (0, 1)$. In contrast, Algorithm 1 reduces to the following when the error condition is assumed to pass after $n$ iterations:

$$\bar{z}_{i+1}^k = (\mathrm{id} + \gamma A)^{-1}(z^k - \gamma F \bar{z}_i^k) \quad \forall i = 0, ..., n-1 \quad \text{with} \quad \bar{z}_0^k = z^k$$
$$z^{k+1} = (1 - \bar{\alpha}_k)z^k + \bar{\alpha}_k(\bar{z}_n^k - \gamma F \bar{z}_n^k + \gamma F \bar{z}_{n-1}^k) \tag{15}$$

where we can guarantee that $\bar{\alpha}_k = \lambda_k \alpha_k \in (0, 2(1 - \sigma + \rho/\gamma))$ since the adaptive stepsize $\alpha_k \in (1 - \sigma + \rho/\gamma, \infty)$ (from Proposition 5.1 by picking $\delta = \rho$) can be absorbed into the relaxation parameter $\lambda_k \in (0, 2)$. Consequently, the convergence guarantee of Theorem 6.1 carries over to (15).

It becomes apparent that for the fixed variant the only difference with RAPP is the additional term $\gamma(F \bar{z}_{n-1}^k - F \bar{z}_n^k)$. This difference is a consequence of using the extragradient evaluation $\bar{v}^k \in \gamma S \bar{z}^k$ in (8) (and consequently Algorithm 1) instead of directly the iterate $\bar{z}^k$ used in RAPP, which is crucial for establishing Fejér monotonicity. Fejér monotonicity is central to avoiding the logarithmic factor in the complexity by circumventing requiring $\sum_{i=0}^{\infty} \|\varepsilon^k\| < \infty$.

For $n = 1$ the fixed variant (15) exactly reduces to CEG+ of Pethick et al. (2022) from which the celebrated forward-backward-forward (FBF) method of Tseng (2000) can be recovered (Pethick et al., 2022, Rem. 3.3). This perspective provides a new interpretation of the FBF algorithm as a one-step application of the (contraction) map used to approximate the proximal operator in (8), akin to the original motivation for the MirrorProx algorithm in Nemirovski (2004). The connection is precise in the sense that we obtain convergence proof of FBF as an instance of Algorithm 1 (cf. Theorem 6.1).

## 7 LINEAR CONVERGENCE

Because of the error correction Algorithm 1 adopts the strong properties of the exact (relaxed) proximal point algorithm (rPPA), even when inexactness is present. Specifically, Fejér monotonicity holds, which we will use to show that linear convergence is automatically obtained under appropriate structure. In contrast, Halpern based methods do not automatically adapt due to the anchoring mechanism and depends on stepsize modifications in order to obtain linear convergence (Park & Ryu, 2022).

We will specifically work under the following assumption.

**Assumption 7.1** (Error bound). *The operator $S = F + A$ satisfies for some $\tau > 0$ that*
$$\|v\| \geq \tau \operatorname{dist}(z, \operatorname{zer} S) \quad \forall(z, v) \in \operatorname{gph} S. \tag{16}$$
*Remark* 7.2. The error bound captures two otherwise seemingly distinct problem classes:

(i) Strongly monotone problems for which $\tau$ is the strong monotonicity modulus.

(ii) Affine operators for which $\tau$ is the minimum non-zero singular value of the linear operator. Nonmonontone affine operators are allowed under the weak MVI (cf. Bauschke et al. (2021, Sec. 5)).

The following theorem establishes linear convergence of Algorithm 1 through the more general implicit scheme (8).

**Theorem 7.3.** *Suppose Assumptions 3.1(iii) and 7.1 hold. Consider the sequence $(z^k)_{k \in \mathbb{N}}$ generated by (8) with $\lambda \in (0, 2)$, $\delta \leq \rho$, $\sigma \in [0, 1 + \frac{\delta}{\gamma})$. Then, for all $z^\star \in \mathcal{Z}^\star$*
$$\operatorname{dist}^2(z^K, \operatorname{zer} S) \leq (1 - \zeta)^K \operatorname{dist}^2(z^0, \operatorname{zer} S)$$
*with $\zeta = \lambda(2 - \lambda)\left(\frac{1 - \sigma + \delta/\gamma}{1 - \sigma + 1/\tau\gamma}\right)^2$.*

*Remark* 7.4. As a by-product, Theorem 7.3 establishes linear convergence for EG+ (Diakonikolas et al., 2021), AdaptiveEG+/CEG+ (Pethick et al., 2022) (when $\rho > -\gamma/2$) and rPPA (Eckstein & Bertsekas, 1992) (when $\rho > -\gamma$). For the reductions see Remark 4.1 and Section 6.1.

## 8 TIGHTNESS

In this section we establish that the derived range of the weak MVI parameter $\rho \in (-\gamma, \infty)$ in Theorems 5.3 and 6.1 is tight for the relaxed proximal point algorithm (rPPA)
$$z^{k+1} = (1 - \bar{\alpha}_k)z^k + \bar{\alpha}_k(\mathrm{id} + \gamma S)^{-1}(z^k) \tag{rPPA}$$

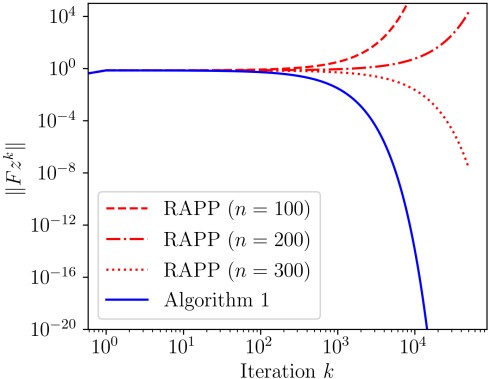 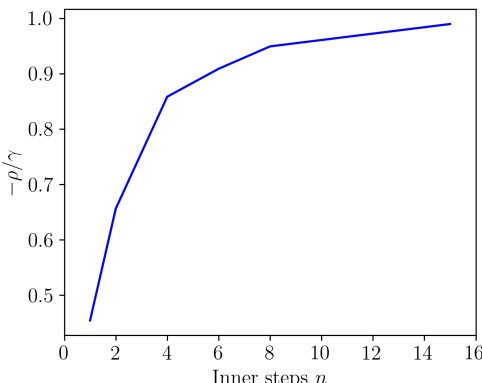

Figure 2: (left) For $\rho = {-0.98}/{L}$, even $n = 200$ inner iteration is not sufficient for RAPP to converge, while Algorithm 1 converges rapidly with only $n = 15$ (adaptively picked). (right) Algorithm 1 can achieve arbitrarily high accuracy for a given $\rho < -\gamma$ using a fixed number of inner steps $n$.

where $(\bar{\alpha}_k)_{k \in \mathbb{N}}$ is a predefined stepsize sequence. Note that rPPA can be seen as a special case of our implicit scheme (8) in the idealized case of exact access to the proximal operator (i.e. $\varepsilon^k = 0$).

**Theorem 8.1.** *Consider a sequence $(z^k)_{k \in \mathbb{N}}$ generated according to rPPA with $\gamma > 0$ and $\bar{\alpha}_k > 0$ for all $k \in \mathbb{N}$. Let $\rho \leq -(1 - \frac{\bar{\alpha}_k}{2})\gamma$. Then, there exists an operator $S : \mathbb{R}^d \to \mathbb{R}^d$ for any $d > 1$ satisfying Assumption 3.1(iii) for which the sequence will not converge.*

*Remark* 8.2. Theorem 8.1 prevents convergence in general when the weak MVI parameter $\rho \leq -\gamma$, even when the relaxation parameter $\bar{\alpha}_k \to 0$. The lower bound $\rho \geq -\gamma/2$ for the (unrelaxed) proximal point algorithm (Gorbunov et al., 2022, Thm. 3.3) is recovered by taking $\bar{\alpha}_k = 1$.

In Appendix G we exploit that the range of $\rho$ is dictated by the stepsize $\gamma$, by analysing a Gauss-Seidel type update rule which permits stepsizes that are possibly larger than $1/L$.

## 9 NUMERICAL EVALUATION

We test Algorithm 1 and RAPP on Pethick et al. (2022, Ex. 5) which can be parameterized by $\rho$ and $L$ (see Example F.1). We set $\gamma = {0.99}/{L}$ for both methods and $\bar{\alpha}_k = 0.001$ for RAPP and compared on $\rho = {-0.98}/{L}$. Algorithm 1 is additionally run on multiple problem instances of varying $\rho$ to determined the relationship with the (automatically selected) number of inner steps. The results are shown in Figure 2, where Algorithm 1 is observed to converge using substantially fewer inner iterations than the baseline.

## 10 CONCLUSION

We have introduced a hybrid proximal extragradient method that interpolates between a relaxed extragradient method and the relaxed proximal point method. The algorithm achieves the first $\mathcal{O}(\frac{1}{\epsilon})$ oracle complexity for the loose requirement $\rho > -1/L$ in weak MVIs, thus removing the logarithmic factor. The algorithm mitigates the need for hyperparameter selection by both automatically determining the stepsize $\alpha_k$ and the number of inner iterations $n$, while automatically enjoying linear convergence under an error bound condition. The construction is obtained through an intuitive geometric interpretation that leads to a tight and simple analysis.

In the special case of cohypomonotone problems our derived rates can be accelerated for $\rho > -1/2L$. It is interesting to investigate integrating the halfspace projection method with Halpern iteration, which could permit an optimal method in the regime $\rho \in \left(-1/L, -1/2L\right]$ without suffering a logarithmic factor in the complexity.

## 11 ACKNOWLEDGEMENTS

This work was supported by the Swiss National Science Foundation (SNSF) under grant number 200021_205011. This work was supported by Hasler Foundation Program: Hasler Responsible AI (project number 21043). Research was sponsored by the Army Research Office and was accomplished under Grant Number W911NF-24-1-0048.

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

# Appendix

TABLE OF CONTENTS

## A  PRELIMINARIES

The distance from $z \in \mathbb{R}^d$ to a set $\mathcal{Z} \subseteq \mathbb{R}^d$ is defined as $\mathrm{dist}(z, \mathcal{Z}) := \min_{z' \in \mathcal{Z}} \|z - z'\|$. The normal cone is defined as $\mathcal{N}_{\mathcal{Z}}(z) := \{\, v \mid \langle v, z' - z \rangle \leq 0 \quad \forall z' \in \mathcal{Z} \,\}$ and the projection as $\mathbf{\Pi}_{\mathcal{Z}}(z) := \arg\min_{w \in \mathcal{Z}} \|z - w\|^2$ and the proximal operator of $g : \mathbb{R}^d \to \mathbb{R}$ as $\mathrm{prox}_{\gamma g}(z) := \arg\min_{w \in \mathcal{Z}} g(w) + \frac{1}{2\gamma}\|z - w\|^2$.

We restate here some common definitions from monotone and nonexpansive operator for convenience (for further details see Bauschke & Combettes (2017)). An operator $A : \mathbb{R}^d \rightrightarrows \mathbb{R}^n$ maps each point $z \in \mathbb{R}^d$ to a subset $Az \subseteq \mathbb{R}^n$, where the notation $A(z)$ and $Az$ will be used interchangably. We denote the domain of $A$ by $\mathrm{dom}A := \{z \in \mathbb{R}^d \mid Az \neq \emptyset\}$, its graph by $\mathrm{gph}\, A := \{(z, v) \in \mathbb{R}^d \times \mathbb{R}^n \mid v \in Az\}$. The inverse of $A$ is defined through its graph, $\mathrm{gph}\, A^{-1} := \{(v, z) \mid (z, v) \in \mathrm{gph}\, A\}$ and the set of its zeros by

$$\mathrm{zer}\, A := \{z \in \mathbb{R}^d \mid 0 \in Az\}.$$

The set of fixed points is defined as

$$\mathrm{fix}\, A := \{z \in \mathbb{R}^d \mid z \in Az\}.$$

**Definition A.1.** *A single-valued operator $T : \mathbb{R}^d \to \mathbb{R}^d$ is said to be*

(i) *nonexpansive if $\|Tz - Tz'\| \leq \|z - z'\| \quad \forall z, z' \in \mathbb{R}^d$.*

(ii) *quasi-nonexpansive if $\|Tz - z^\star\| \leq \|z - z^\star\| \quad (\forall z \in \mathbb{R}^d)(\forall z^\star \in \mathrm{fix}\, T)$.*

(iii) *firmly nonexpansive if $\|Tz - Tz'\|^2 \leq \|z - z'\|^2 - \|(z - z') - (Tz - Tz')\|^2 \quad \forall z, z' \in \mathbb{R}^d$.*

(iv) *firmly quasi-nonexpansive if $\|Tz - z^\star\|^2 \leq \|z - z^\star\|^2 - \|z - Tz\|^2 \quad (\forall z \in \mathbb{R}^d)(\forall z^\star \in \mathrm{fix}\, T)$.*

The resolvent operator $J_A := (\mathrm{id} + A)^{-1}$ is firmly nonexpansive (with $\mathrm{dom}J_A = \mathbb{R}^d$) iff $A$ is maximally monotone.

Let $\mathcal{S} \subseteq \mathbb{R}^d$. A sequence $(z^k)_{k \in \mathbb{N}}$ is said to be

(i) Fejér monotone if $\|z^{k+1} - z\| \leq \|z^k - z\| \quad (\forall z \in \mathcal{S})(\forall k \in \mathbb{N})$

(ii) Quasi-Fejér monotone if $\|z^{k+1} - z\|^2 \leq \|z^k - z\|^2 + e_k \quad (\forall z \in \mathcal{S})(\forall k \in \mathbb{N})$

where $(e_k)_{k \in \mathbb{N}}$ is a summable sequence in $(0, \infty)$.

**Definition A.2** (monotonicity Bauschke & Combettes (2017)). *An operator $A : \mathbb{R}^d \rightrightarrows \mathbb{R}^d$ is called monotone if,*

$$\langle v - v', z - z' \rangle \geq 0 \quad \forall (z, v), (z', v') \in \mathrm{gph}\, A,$$

*The operator $A$ is* maximally *monotone if no other monotone operator $B$ exists for which $\mathrm{gph}\, A \subset \mathrm{gph}\, B$.*

**Definition A.3** (Lipschitz continuity and cocoercivity). *Let $\mathcal{D} \subseteq \mathbb{R}^d$ be a nonempty set. A single-valued operator $A : \mathcal{D} \to \mathbb{R}^d$ is said to be L-Lipschitz continuous if for any $z, z' \in \mathcal{D}$*

$$\|Az - Az'\| \leq L\|z - z'\|,$$

*and $\beta$-cocoercive if*

$$\langle z - z', Az - Az' \rangle \geq \beta\|Az - Az'\|^2.$$

**Lemma A.4.** *Let $A : \mathbb{R}^d \to \mathbb{R}^d$ denote a single valued operator. Then,*

(i) *$A$ is 1-Lipschitz if and only if $T = \mathrm{id} - A$ is $1/2$-cocoercive.*

(ii) *If $A$ is L-Lipschitz, then $T = \mathrm{id} - \eta A$, $\eta \in (0, 1/L)$, is $(1 - \eta L)$-monotone, and in particular for all $u, v \in \mathbb{R}^d$,*

$$\|Tu - Tv\| \geq (1 - \eta L)\|u - v\|. \tag{17}$$

*Proof.* The first claim follows directly from (Bauschke & Combettes, 2017, Prop.4.11). That $T$ is strongly monotone is a consequence of the Cauchy Schwarz inequality and Lipschitz continuity of $A$:

$$\langle Tv - Tu, v - u \rangle = \|v - u\|^2 - \eta \langle Av - Au, v - u \rangle \geq (1 - \eta L)\|v - u\|^2.$$

In turn, the last claim follows from the Cauchy-Schwarz inequality. $\qquad\square$

We will use the following closed form solution concerning hyperplane projections.

**Fact A.5.** *The projection* $\mathbf{\Pi}_{\mathcal{D}}(x) := \arg\min_{z \in \mathcal{D}} \|z - x\|^2$ *onto the set* $\mathcal{D} = \{z \mid \langle a, z \rangle \geq b\}$ *of* $x \notin \mathcal{D}$ *is given as,*

$$\mathbf{\Pi}_{\mathcal{D}}(x) = x - \frac{\langle a, x \rangle - b}{\|a\|^2} a. \tag{18}$$

## B    RECOVERING EXISTING SCHEMES

Let us recall the proposed implicit scheme in (8), which for $\gamma > 0$, $\lambda_k \in (0, 2)$, $\delta \leq \rho$ and $\sigma \in (0, 1 + \frac{\delta}{\gamma})$, proceeds as follows:

$$
\begin{aligned}
\text{find} \quad & \bar{z}^k \in \mathbb{R}^d \quad \text{and} \quad \bar{v}^k \in \gamma S \bar{z}^k \\
\text{s.t.} \quad & \bar{z}^k = z^k - (\bar{v}^k + \varepsilon^k) \quad \text{and} \quad -\langle \varepsilon^k, \bar{v}^k \rangle \leq \sigma \|\bar{v}^k\|^2 \\
\text{update} \quad & z^{k+1} = z^k - \lambda_k \alpha_k \bar{v}^k \quad \alpha_k = \frac{\langle \bar{v}^k, z^k - \bar{z}^k \rangle}{\|\bar{v}^k\|^2} + \frac{\delta}{\gamma}
\end{aligned}
\tag{19}
$$

The scheme can recover a range of existing methods.

**The relaxed proximal point algorithm**    Take the error to be $\varepsilon^k = 0$ in which case $\alpha_k = 1 + \frac{\delta}{\gamma}$ and the error condition is satisfied for any $\sigma$. By using that $\bar{v}^k = z^k - \bar{z}^k$, the update (19) reduces to the relaxed proximal point update (Eckstein & Bertsekas, 1992)

$$
\begin{aligned}
\bar{z}^k &= (\text{id} + \gamma S)^{-1}(z^k) \\
z^{k+1} &= (1 - \lambda_k)z^k + \lambda_k \bar{z}^k
\end{aligned}
\tag{20}
$$

where $\gamma > 0$, $\lambda_k \in (0, 2(1 + \frac{\delta}{\gamma}))$ and $\delta \leq \rho$.

**Solodov & Svaiter**    In the monotone case ($\rho = 0$) and with the more stringent error conditioning $\|\varepsilon^k\| \leq \sigma \max\{\|\bar{v}^k\|, \|z^k - \bar{z}^k\|\}$, (19) reduces to the method of Solodov & Svaiter (1999a)

$$
\begin{aligned}
\text{find} \quad & \bar{z}^k \in \mathbb{R}^d \quad \text{and} \quad \bar{v}^k \in \gamma S \bar{z}^k \\
\text{s.t.} \quad & \bar{z}^k = z^k - (\bar{v}^k + \varepsilon^k) \quad \text{and} \quad \|\varepsilon^k\| \leq \sigma \max\{\|\bar{v}^k\|, \|z^k - \bar{z}^k\|\} \\
\text{update} \quad & z^{k+1} = z^k - \lambda_k \alpha_k \bar{v}^k \quad \alpha_k = \frac{\langle \bar{v}^k, z^k - \bar{z}^k \rangle}{\|\bar{v}^k\|^2}
\end{aligned}
\tag{21}
$$

for $\gamma > 0$, $\lambda_k \in (0, 2)$ and $\sigma \in [0, 1)$.

**AdaptiveEG+**    Pick $\varepsilon^k = Fz^k - F\bar{z}^k$. Notice that the error is a natural choice as it replaces the implicit evaluation $F\bar{z}^k$ in $\bar{v}^k$ with the known $Fz^k$. Define the forward operator as $H := \text{id} - \gamma F$ such that $\bar{v}^k = Hz^k - H\bar{z}^k$. Then (19) reduces to the scheme of Pethick et al. (2023b):

$$
\begin{aligned}
\bar{z}^k &= (\text{id} + \gamma A)^{-1}(Hz^k) \\
z^{k+1} &= z^k - \lambda_k \alpha_k (Hz^k - H\bar{z}^k) \quad \alpha_k = \frac{\langle Hz^k - H\bar{z}^k, z^k - \bar{z}^k \rangle}{\|Hz^k - H\bar{z}^k\|^2} + \frac{\delta}{\gamma}
\end{aligned}
\tag{22}
$$

where $\gamma \in (\lfloor -2\rho \rfloor_+, 1/L)$, $\lambda_k \in (0, 2)$ and $\delta \in (-\gamma/2, \rho]$. A constant stepsize variant can be obtained by absorbing the adaptive $\alpha_k$ into $\lambda_k$. Let $\lambda = \lambda_k \alpha_k$, then the resulting scheme can be written as

$$
\begin{aligned}
\bar{z}^k &= (\text{id} + \gamma A)^{-1}(Hz^k) \\
z^{k+1} &= z^k - \lambda(Hz^k - H\bar{z}^k)
\end{aligned}
\tag{23}
$$

where $\gamma \in (\lfloor -2\rho \rfloor_+, 1/L]$, $\lambda \in (0, 1 + \frac{2\delta}{\gamma})$ and $\delta \in (-\gamma/2, \rho]$. When in the unconstrained case ($A \equiv 0$) the scheme reduces to the extragradient+ method (Diakonikolas et al., 2021)

$$
\begin{aligned}
\bar{z}^k &= z^k - \gamma F z^k \\
z^{k+1} &= z^k - \lambda \gamma F \bar{z}^k
\end{aligned}
\tag{EG+}
$$

which we synonymously refer to as the relaxed extragradient method since EG+ can be rewritten as

$$
\begin{aligned}
\bar{z}^k &= z^k - \gamma F z^k \\
z^{k+1} &= (1-\lambda)z^k + \lambda(z^k - \gamma F \bar{z}^k)
\end{aligned}
\tag{24}
$$

**Forward-backward-forward**  In the monotone case ($\rho = 0$) we can recover the celebrated forward-backward-forward (FBF) method of Tseng (Tseng, 2000). Pick $\varepsilon^k = F z^k - F \bar{z}^k$ and $\gamma < 1/L$ (strictly). This allows us to pick $\lambda_k \alpha_k = 1$ as noted in Pethick et al. (2023b, Rem. 3.3) in which case the scheme simplifies to

$$
\begin{aligned}
\bar{z}^k &= (\mathrm{id} + \gamma A)^{-1}(z^k - \gamma F z^k) \\
z^{k+1} &= \bar{z}^k - \gamma(F \bar{z}^k - F z^k)
\end{aligned}
\tag{FBF}
$$

with $\gamma \in (0, 1/L)$. When in the unconstrained case ($A \equiv 0$) the scheme reduces to the extragradient method (Korpelevich, 1977)

$$
\begin{aligned}
\bar{z}^k &= z^k - \gamma F z^k \\
z^{k+1} &= z^k - \gamma F \bar{z}^k
\end{aligned}
\tag{EG}
$$

## C  Proofs for Section 5 (Analysis of the implicit scheme (8))

**Proposition 5.1** (Properties of (7))**.** *Suppose Assumption 3.1(iii) holds, $\delta \leq \rho$, and (5) satisfies the following error condition,*

$$
-\langle \varepsilon, \bar{v} \rangle \leq \sigma \|\bar{v}\|^2.
\tag{12}
$$

*where $\sigma \in [0, 1 + \frac{\delta}{\gamma})$. Then,*

(i)  *The projection operator $P : \mathbb{R}^d \to \mathbb{R}^d$ in (7) is firmly quasi-nonexpansive.*

(ii)  $\mathcal{Z}^\star \subseteq \mathrm{fix}\, P \subseteq \mathrm{zer}\, S$.

(iii)  *The closed form solution to $P$ is given as in (7) and the stepsize satisfies $\alpha \geq 1 + \frac{\delta}{\gamma} - \sigma$.*

*Proof.* To find the closed form solution for the projection we invoke Fact A.5, concerning general hyperplane projections, with $a = -\bar{v}$ and $b = \langle \bar{v}, \bar{z} \rangle - \frac{\delta}{\gamma}\|\bar{v}\|^2$. For $u \notin \mathcal{D}(z)$,

$$
\mathbf{\Pi}_{\mathcal{D}(z)}(u) = u - \frac{\langle \bar{v}, u - \bar{z} \rangle + \frac{\delta}{\gamma}\|\bar{v}\|^2}{\|\bar{v}\|^2} \bar{v}
\tag{25}
$$

Let us prove $\mathcal{Z}^\star \subseteq \mathrm{fix}\, P$. The set $\mathcal{D}(z)$ defined in (6) is constructed to contain the solution set. Let us verify this claim. From the definition of $\bar{z}$ in (5) we have that $\bar{v} \in \gamma S \bar{z}$. By using Assumption 3.1(iii), a solution $z^\star \in \mathcal{Z}^\star$ satisfies

$$
\langle \bar{v}, \bar{z} - z^\star \rangle \geq \frac{\rho}{\gamma}\|\bar{v}\|^2,
$$

which is contained in $\mathcal{D}(z)$ by assuming $\delta \leq \rho$. This proofs that $\mathcal{Z}^\star \subseteq \mathcal{D}(z)$ for all $z \in \mathbb{R}^d$, which is more general than $\mathcal{Z}^\star \subseteq \mathrm{fix}\, P$.

To prove $\mathrm{fix}\, P \subseteq \mathrm{zer}\, S$ we need to show that the (adaptive) stepsize in (25) is positive and bounded away from zero. We have

$$
\langle \bar{v}, z - \bar{z} \rangle + \frac{\delta}{\gamma}\|\bar{v}\|^2 = (1 + \frac{\delta}{\gamma})\|\bar{v}\|^2 + \langle \varepsilon, \bar{v} \rangle \geq (1 + \frac{\delta}{\gamma} - \sigma)\|\bar{v}\|^2
\tag{26}
$$

where the third last inequality follows from assuming $-\langle \varepsilon, \bar{v} \rangle \leq \sigma\|\bar{v}\|^2$. Thus, (26) is strictly positive assuming $\sigma < 1 + \frac{\delta}{\gamma}$ (which proofs the last claim). Consequently, $z \in \mathrm{fix}\, P$ only if $0 = \bar{v} \in S\bar{z}$. This proves the second claim.

Finally for the first claim, the projection onto a convex set is firmly quasi-nonexpansive. That is, for all $z \in \mathbb{R}^d$ and $z^\star \in \mathcal{Z}^\star \subseteq \text{fix}\,\mathbf{\Pi}_{\mathcal{D}(z)}$

$$\|\mathbf{\Pi}_{\mathcal{D}(z)}(u) - z^\star\|^2 \leq \|u - z^\star\|^2 - \|u - \mathbf{\Pi}_{\mathcal{D}(z)}(u)\|^2$$

where we can take $u = z$ to establish the claim. This completes the proof. □

For completeness we include a well known convergence guarantee for the KM iterations applied to a firmly quasi-nonexpansive operator, which is used to establish convergence of (8).

**Theorem C.1.** *Suppose $T : \mathbb{R}^d \to \mathbb{R}^d$ is firmly quasi-nonexpansive. Consider the sequence $(z^k)_{k \in \mathbb{N}}$ generated by KM with $\lambda_k \in (0, 2)$. Then, for all $z^\star \in \text{fix}\,T$*

$$\min_{k \in \{0,\dots,K-1\}} \|Tz^k - z^k\|^2 \leq \frac{\|z^0 - z^\star\|^2}{\lambda_k(2 - \lambda_k)K}.$$

*Proof.* We have

$$\begin{aligned}
\|z^{k+1} - z^\star\|^2 &= (1 - \lambda_k)\|z^k - z^\star\|^2 + \lambda_k\|Tz^k - z^\star\|^2 - \lambda_k(1 - \lambda_k)\|Tz^k - z^k\|^2 \\
&= (1 - \lambda_k)\|z^k - z^\star\|^2 + \lambda_k\|Tz^k - Tz^\star\|^2 - \lambda_k(1 - \lambda_k)\|Tz^k - z^k\|^2 \\
&\leq (1 - \lambda_k)\|z^k - z^\star\|^2 + \lambda_k\|z^k - z^\star\|^2 - \lambda_k\|Tz^k - z^k\|^2 - \lambda_k(1 - \lambda_k)\|Tz^k - z^k\|^2 \\
&= \|z^k - z^\star\|^2 - \lambda_k(2 - \lambda_k)\|Tz^k - z^k\|^2. \quad\quad\quad (27)
\end{aligned}$$

where we have used firmly quasi-nonexpansiveness of $T$. Telescoping completes the proof. □

**Theorem 5.3.** *Suppose Assumption 3.1(iii) holds. Consider the sequence $(z^k)_{k \in \mathbb{N}}$ generated by (8) with $\lambda_k \in (0, 2)$, $\kappa := \liminf_{k\to\infty} \lambda_k(2 - \lambda_k) > 0$, $\delta \leq \rho$, and $\sigma \in [0, 1 + \frac{\delta}{\gamma})$. Then, for all $z^\star \in \mathcal{Z}^\star$*

$$\min_{k \in \{0,\dots,K-1\}} \text{dist}(0, S\bar{z}^k)^2 \leq \frac{\|z^0 - z^\star\|^2}{\kappa\gamma^2(1 + \frac{\delta}{\gamma} - \sigma)^2 K}.$$

*Furthermore, if $\mathcal{Z}^\star = \text{zer}\,S$ then $(z^k)_{k \in \mathbb{N}}$ converges to some $z^\star \in \mathcal{Z}^\star$.*

*Proof.* From Proposition 5.1 and Theorem C.1 we have that

$$\min_{k \in \{0,\dots,K-1\}} \lambda_k(2 - \lambda_k)\|\alpha_k \bar{v}^k\|^2 \leq \frac{\|z^0 - z^\star\|^2}{K}.$$

From Proposition 5.1 we have that $\alpha_k \geq (1 + \frac{\delta}{\gamma} - \sigma)$. Lower bounding $\alpha_k$ in the convergence guarantee and rearranging obtains the claimed rate.

It follows from Fejér monotonicity with respect to fix $P$ that $(z^k)_{k \in \mathbb{N}}$ is bounded. Using a telescoping argument we have that $(\|\bar{v}^k\|)_{k \in \mathbb{N}}$ converges to zero, implying that the limit points of $z^k$ belong to zer $S$ when zer $S = \mathcal{Z}^\star$ as claimed. □

**Lemma C.2.** *Suppose Assumptions 3.1(i) and 3.1(ii) hold and the stepsize $\gamma < \frac{1}{L}$. Then the resolvent $J_{\gamma S}$ is single-valued and has full domain.*

*Proof.* The operator $\gamma S$ is $-\gamma L$-monotone due to Assumptions 3.1(i) and 3.1(ii). Consequently $(\gamma S)^{-1}$ is $-\gamma L$-comonotone since $\gamma < \frac{1}{L}$ due to Bauschke et al. (2021, Lem. 2.8). Thus, the resolvent $J_{\gamma S}$ is single-valued and has full domain due to Bauschke et al. (2021, Cor. 2.14) which completes the proof. □

# D   PROOFS FOR SECTION 6 (ANALYSIS OF THE EXPLICIT SCHEME (ALGORITHM 1))

**Theorem 6.1.** *Suppose Assumption 3.1 holds. Consider the sequence $(z^k)_{k \in \mathbb{N}}$ generated by Algorithm 1 and let $\kappa := \liminf_{k\to\infty} \lambda_k(2 - \lambda_k) > 0$. Then, for all $z^\star \in \mathcal{Z}^\star$*

$$\min_{k \in \{0,\dots,K-1\}} \text{dist}(0, S\bar{z}^k)^2 \leq \frac{\|z^0 - z^\star\|^2}{\kappa\gamma^2(1 + \frac{\delta}{\gamma} - \sigma)^2 K}.$$

*Furthermore, if $\mathcal{Z}^\star = \text{zer } S$ then $(z^k)_{k\in\mathbb{N}}$ converges to some $z^\star \in \mathcal{Z}^\star$. Moreover, let $n$ denote the number of inner iterations in Algorithm 1. Then,*

    (i) *the error condition passes immediately (i.e. $n = 1$) even when only $\gamma \le \frac{1}{L}$ if $\sigma \in (\frac{1}{2}, 1 + \frac{\rho}{\gamma})$ which is feasible for $\rho > -\frac{\gamma}{2}$.*

    (ii) *the error condition passes after at most $n$ iterations for some $\gamma < {}^1\!/_L$ if $\sigma \in (\frac{2 - \ln(n)/n}{n+1 - \ln(n)/n}, 1 + \frac{\rho}{\gamma})$ which is feasible for $\rho > -\frac{1}{L}(1 - \frac{\ln(n)}{n})\frac{n-1}{n+1-\ln(n)/n}$.*

*Proof.* The convergence rate and convergence of $(z^k)_{k\in\mathbb{N}}$ follows immediately from Theorem 5.3. The resolvent $J_{\gamma A}$ is single-valued and has full domain due to maximally monotonicity of the operator $A$. Theorem 6.1(i) follows from the argument in Section 6.1. Theorem 6.1(ii) follows directly from Lemma D.7 due to Assumptions 3.1(i) and 3.1(ii). $\qquad\square$

**Corollary 6.3.** *Suppose Assumption 3.1 holds. Consider the sequence $(z^k)_{k\in\mathbb{N}}$ generated by Algorithm 1 and let $\kappa := \liminf_{k\to\infty} \lambda_k(2 - \lambda_k) > 0$. Then, for all $z^\star \in \mathcal{Z}^\star$ the sequence achieves $\min_{k\in\{0,\ldots,K-1\}} \text{dist}(0, S\bar{z}^k)^2 \le \epsilon$ after at most*

$$\#(\text{oracle calls}) \le \frac{(1+n)\|z^0 - z^\star\|^2}{\kappa\gamma^2(1 + \frac{\delta}{\gamma} - \sigma)^2 \epsilon}$$

*to both the operator $F$ and the resolvent $(\text{id} + \gamma A)^{-1}$ where $n = \lceil \frac{6}{1+\rho L} \ln(\frac{3}{1+\rho L}) \rceil$ is an upper bound on the number of inner iterations.*

*Proof.* Due to Assumptions 3.1(i) and 3.1(ii), from Lemma D.8 we know that $n = \lceil \frac{6}{1+\rho L} \ln(\frac{3}{1+\rho L}) \rceil$ inner iterations are sufficient for satisfying the error condition. Note that one additional evaluation of $F$ is carried out at each iteration, i.e. $F\bar{z}^k$. Combined with the rate in Theorem 6.1, the claim is obtained. $\qquad\square$

## D.1 Analysis of the inner iterations

In this section we provide a formal proof for the lower bound on the value of $\rho$. We have that:

$$\sigma \le \rho/\gamma + 1 \quad \Leftrightarrow \quad \rho \ge -\gamma(1 - \sigma)$$

So we can start by providing a lower bound on the value of $\sigma$.

**Notation:** We will index the inner iterations of Algorithm 1 by $j$, meaning that we can rewrite the $j^{\text{th}}$ iteration of the inner loop, as:

$$\begin{aligned} h_j^k &= z^k - \gamma F\bar{z}_{j-1}^k \\ \bar{z}_j^k &= (\text{id} + \gamma A)^{-1} h_j^k \end{aligned}$$

For the sake of exposition, let us rewrite the condition of line 5 of Algorithm 1, using this notation:

$$\langle z^k - \bar{z}_j^k, \bar{v}_j^k \rangle \ge (1 - \sigma)\left\|\bar{v}_j^k\right\|^2, \text{ where } \bar{v}_j^k = h_j^k - \bar{z}_j^k + \gamma F\bar{z}_j^k \tag{28}$$

### D.1.1 Lower bound on $\sigma$

If $\left\|\bar{v}_j^k\right\| = 0$, then (28) holds trivially for any $\sigma$, so for the rest of this section, we will analyse the case where $\left\|\bar{v}_j^k\right\| > 0$

**Lemma D.1.** *For $j$ inner iterations of Algorithm 1, we have that:*

$$\sigma \ge \frac{\left\|\gamma F\bar{z}_j^k - \gamma F\bar{z}_{j-1}^k\right\|}{\left\|z^k - \bar{z}_j^k\right\| + \left\|\gamma F\bar{z}_j^k - \gamma F\bar{z}_{j-1}^k\right\|}$$

*Proof.* We have that:

$$
\begin{aligned}
z^k - \bar{z}_j^k &= z^k - \gamma F \bar{z}_{j-1}^k - \bar{z}_j^k + \gamma F \bar{z}_{j-1}^k \\
&= h_j^k - \bar{z}_j^k + \gamma F \bar{z}_{j-1}^k
\end{aligned}
$$

Let us now analyze the left hand side of (28).

$$
\begin{aligned}
2\langle z^k - \bar{z}_j^k, \bar{v}_j^k \rangle &= \left\| z^k - \bar{z}_j^k \right\|^2 + \left\| \bar{v}_j^k \right\|^2 - \left\| z^k - \bar{z}_j^k - \bar{v}_j^k \right\|^2 \\
&= \left\| z^k - \bar{z}_j^k \right\|^2 + \left\| \bar{v}_j^k \right\|^2 - \left\| \gamma F \bar{z}_{j-1}^k - \gamma F \bar{z}_j^k \right\|^2
\end{aligned}
$$

So from (28), we have the following bound for $\sigma$

$$
\begin{aligned}
2 - 2\sigma &\leq 1 + \frac{\left\| z^k - \bar{z}_j^k \right\|^2 - \left\| \gamma F \bar{z}_{j-1}^k - \gamma F \bar{z}_j^k \right\|^2}{\left\| \bar{v}_j^k \right\|^2} \\
\Leftrightarrow \quad \sigma &\geq \frac{1}{2}\left( 1 - \frac{\left\| z^k - \bar{z}_j^k \right\|^2 - \left\| \gamma F \bar{z}_{j-1}^k - \gamma F \bar{z}_j^k \right\|^2}{\left\| \bar{v}_j^k \right\|^2} \right)
\end{aligned}
$$

We will now maximize the above lower bound of $\sigma$, which is equivalen to maximizing for $\left\| \bar{v}_j^k \right\|^2$. Since $\bar{v}_j^k = z^k - \bar{z}_j^k + \gamma F \bar{z}_j^k - \gamma F \bar{z}_{j-1}^k$, we get from the triangular inequality, that:

$$
\left\| \bar{v}_j^k \right\| \leq \left\| z^k - \bar{z}_j^k \right\| + \left\| \gamma F \bar{z}_j^k - \gamma F \bar{z}_{j-1}^k \right\|
$$

So we get:

$$
\begin{aligned}
\sigma &\geq \frac{1}{2}\left( 1 - \frac{\left\| z^k - \bar{z}_j^k \right\|^2 - \left\| \gamma F \bar{z}_{j-1}^k - \gamma F \bar{z}_j^k \right\|^2}{\left( \left\| z^k - \bar{z}_j^k \right\| + \left\| \gamma F \bar{z}_j^k - \gamma F \bar{z}_{j-1}^k \right\| \right)^2} \right) \\
\Leftrightarrow \quad \sigma &\geq \frac{1}{2}\left( 1 - \frac{\left( \left\| z^k - \bar{z}_j^k \right\| - \left\| \gamma F \bar{z}_{j-1}^k - \gamma F \bar{z}_j^k \right\| \right)\left( \left\| z^k - \bar{z}_j^k \right\| + \left\| \gamma F \bar{z}_{j-1}^k - \gamma F \bar{z}_j^k \right\| \right)}{\left( \left\| z^k - \bar{z}_j^k \right\| + \left\| \gamma F \bar{z}_j^k - \gamma F \bar{z}_{j-1}^k \right\| \right)^2} \right) \\
\Leftrightarrow \quad \sigma &\geq \frac{1}{2}\left( 1 - \frac{\left\| z^k - \bar{z}_j^k \right\| - \left\| \gamma F \bar{z}_{j-1}^k - \gamma F \bar{z}_j^k \right\|}{\left\| z^k - \bar{z}_j^k \right\| + \left\| \gamma F \bar{z}_j^k - \gamma F \bar{z}_{j-1}^k \right\|} \right) \\
\Leftrightarrow \quad \sigma &\geq \frac{1}{2}\left( \frac{\left\| z^k - \bar{z}_j^k \right\| + \left\| \gamma F \bar{z}_j^k - \gamma F \bar{z}_{j-1}^k \right\| - \left\| z^k - \bar{z}_j^k \right\| + \left\| \gamma F \bar{z}_{j-1}^k - \gamma F \bar{z}_j^k \right\|}{\left\| z^k - \bar{z}_j^k \right\| + \left\| \gamma F \bar{z}_j^k - \gamma F \bar{z}_{j-1}^k \right\|} \right) \\
\Leftrightarrow \quad \sigma &\geq \frac{\left\| \gamma F \bar{z}_j^k - \gamma F \bar{z}_{j-1}^k \right\|}{\left\| z^k - \bar{z}_j^k \right\| + \left\| \gamma F \bar{z}_j^k - \gamma F \bar{z}_{j-1}^k \right\|}
\end{aligned}
$$

$\square$

### D.1.2 CONTRACTION BOUNDS

**Lemma D.2.** *Suppose Assumptions 3.1(i) and 3.1(ii). For stepsize $\gamma$ and $j$ inner iterations, we have:*

$$
\left\| \bar{z}_j^k - \bar{z}_{j+1}^k \right\| \leq (\gamma L)^j \left\| z^k - \bar{z}_1^k \right\|
$$

*Proof.* For the $j$ inner iteration, we have that:

$$
\begin{aligned}
\left\| \bar{z}_j^k - \bar{z}_{j+1}^k \right\| &= \left\| (\mathrm{id} + \gamma A)^{-1} h_j^k - (\mathrm{id} + \gamma A)^{-1} h_{j+1}^k \right\| \\
\text{(Assumption 3.1(ii))} \quad &\leq \left\| h_j^k - h_{j+1}^k \right\| \\
&= \left\| \gamma F \bar{z}_{j-1}^k - \gamma F \bar{z}_j^k \right\| \\
\text{(Assumption 3.1(i))} \quad &\leq \gamma L \left\| \bar{z}_{j-1}^k - \bar{z}_j^k \right\|
\end{aligned}
$$

By doing this recursively, we get the lemma statement. $\square$

**Lemma D.3.** *Suppose Assumptions 3.1(i) and 3.1(ii). For stepsize $\gamma$ and $j$ inner iterations, we have:*

$$\left\| \gamma F \bar{z}_j^k - \gamma F \bar{z}_{j+1}^k \right\| \leq (\gamma L)^{j+1} \left\| z^k - \bar{z}_1^k \right\|$$

*Proof.*

$$\left\| \gamma F \bar{z}_j^k - \gamma F \bar{z}_{j+1}^k \right\| \leq (\gamma L) \left\| \bar{z}_j^k - \bar{z}_{j+1}^k \right\|$$

The statement follows from the previous lemma. $\qquad\square$

### D.1.3 Lower bound for $\left\| z_k - \bar{z}_j^k \right\|$

**Lemma D.4.** *Suppose Assumptions 3.1(i) and 3.1(ii). For $j$ inner iterations, we have the following inequality:*

$$\left\| z^k - \bar{z}_{j+1}^k \right\| \geq \beta \left\| z^k - \bar{z}_1^k \right\| \tag{29}$$

*where $\beta = \frac{1-(\gamma L)^{j+1}}{1+\gamma L}$*

*Proof.* We will proceed by proof by contraction. Let us assume that for $j$ inner iterations we have the following inequality

$$\left\| z^k - \bar{z}_{j+1}^k \right\| < \beta \left\| z^k - \bar{z}_1^k \right\| \tag{30}$$

for $\beta = \frac{1-(\gamma L)^{j+1}}{1+\gamma L}$.
We have that:

$$\left\| h_1^k - h_{j+1}^k \right\| = \left\| z^k - \gamma F \bar{z}_{j+1}^k - z^k + \gamma F z^k \right\| = \left\| \gamma F \bar{z}_{j+1}^k - \gamma F z^k \right\|$$

Note that in Algorithm 1, we set $\bar{z}_0^k = z^k$. Additionally, we have that: $h_{j+1}^k + \gamma F \bar{z}_j^k = z^k$, for all $j \geq 0$.
Then we have that:

$$
\begin{aligned}
\left\| h_1^k - h_{j+1}^k - (\mathrm{id} + \gamma A)^{-1} h_1^k \right. & - \left. (\mathrm{id} + \gamma A)^{-1} h_{j+1}^k + \gamma F z^k - \gamma F \bar{z}_j^k \right\| \\
&= \left\| z^k - z^k - (\mathrm{id} + \gamma A)^{-1} h_1^k - (\mathrm{id} + \gamma A)^{-1} h_{j+1}^k \right\| \\
\text{(Assumption 3.1(ii))} \quad &\leq \left\| h_1^k - h_{j+1}^k \right\| \\
&= \left\| \gamma F z^k - \gamma F \bar{z}_j^k \right\| \\
\text{(Assumption 3.1(i))} \quad &\leq (\gamma L) \left\| z^k - \bar{z}_j^k \right\|
\end{aligned}
$$

The first inequality comes from (firmly) nonexpansiveness of the resolvent when the operator is maximally monotone, while the second one comes from the Lipschitz of the operator $F$. Now applying the triangle inequality, we get:

$$
\begin{aligned}
\left\| h_1^k - h_{j+1}^k - (\mathrm{id} + \gamma A)^{-1} h_1^k \right. & - \left. (\mathrm{id} + \gamma A)^{-1} h_{j+1}^k + \gamma F z^k - \gamma F \bar{z}_j^k \right\| \\
&\geq \left\| h_1^k - (\mathrm{id} + \gamma A)^{-1} h_1^k + \gamma F z^k \right\| \\
&\quad - \left\| h_{j+1}^k - (\mathrm{id} + \gamma A)^{-1} h_{j+1}^k + \gamma F \bar{z}_j^k \right\| \\
&> (1 - \beta) \left\| z^k - \bar{z}_1^k \right\|
\end{aligned}
$$

Above we used the assumption of the proof (30). Combining the two previous inequality, we get:

$$\left\| z^k - \bar{z}_j^k \right\| > \frac{1-\beta}{\gamma L} \left\| z^k - \bar{z}_1^k \right\| \tag{31}$$

From the triangle inequality, we have that:

$$
\begin{aligned}
\left\| z^k - \bar{z}_j^k \right\| - \left\| z^k - \bar{z}_{j+1}^k \right\| &\leq \left\| \bar{z}_j^k - \bar{z}_{j+1}^k \right\| &\Rightarrow \\
\left\| z^k - \bar{z}_j^k \right\| - \left\| z^k - \bar{z}_{j+1}^k \right\| &\leq (\gamma L)^j \left\| z^k - \bar{z}_1^k \right\| &\Rightarrow \\
\frac{1-\beta}{\gamma L} \left\| z^k - \bar{z}_1^k \right\| - \beta \left\| z^k - \bar{z}_1^k \right\| &< (\gamma L)^j \left\| z^k - \bar{z}_1^k \right\| &\Rightarrow \\
\beta &> \frac{1-(\gamma L)^{j+1}}{1+\gamma L}
\end{aligned}
$$

Where the second line comes from Lemma D.2. The third line comes from our assumption in the proof, (30) and (31). Thus our assumption (30) is violated which concludes the proof. $\qquad\square$

### D.1.4 Proof of Lemma D.6

In this section we will provide the bounds on $\sigma$ and $\rho$ as a function of the product $\gamma L$ and the inner iterations $j$ of Algorithm 1.

**Lemma D.5.** *Suppose Assumptions 3.1(i) and 3.1(ii). For $j$ iterations of the inner loop of Algorithm 1 and stepsize $\gamma < 1/L$, we have that the error condition of line 5 of Algorithm 1 has been satisfied, for*

$$\sigma \geq \frac{(\gamma L)^j + (\gamma L)^{j+1}}{1 + (\gamma L)^{j+1}} \tag{32}$$

*Proof.* From Lemma D.1, we have that:

$$\sigma \geq \frac{\left\|\gamma F \bar{z}_j^k - \gamma F \bar{z}_{j-1}^k\right\|}{\left\|z^k - \bar{z}_j^k\right\| + \left\|\gamma F \bar{z}_j^k - \gamma F \bar{z}_{j-1}^k\right\|}$$

in order to get the tightest lower bound on $\sigma$ we want to maximize the right hand side of this inequality. We can do this by minimizing the value of $\left\|z^k - \bar{z}_j^k\right\|$ and maximizing the value of $\left\|\gamma F \bar{z}_j^k - \gamma F \bar{z}_{j-1}^k\right\|$.
From Lemma D.4, we get that:

$$\left\|z^k - \bar{z}_j^k\right\| \geq \beta \left\|z^k - \bar{z}_1^k\right\|$$

where $\beta = \frac{1 - (\gamma L)^j}{1 + \gamma L}$
and from Lemma D.3

$$\left\|\gamma F \bar{z}_j^k - \gamma F \bar{z}_j^k\right\| \leq (\gamma L)^j \left\|z^k - \bar{z}_1^k\right\|$$

combining the above, we get that:

$$\begin{aligned}
\sigma &\geq \frac{(\gamma L)^j \left\|z^k - \bar{z}_1^k\right\|}{\frac{1 - (\gamma L)^j}{1 + \gamma L} \left\|z^k - \bar{z}_1^k\right\| + (\gamma L)^j \left\|z^k - \bar{z}_1^k\right\|} \\
&= \frac{(\gamma L)^j}{\frac{1 - (\gamma L)^j}{1 + \gamma L} + (\gamma L)^j} \\
&\geq \frac{(\gamma L)^j + (\gamma L)^{j+1}}{1 + (\gamma L)^{j+1}}
\end{aligned}$$

$\qquad\square$

Now using Lemma D.5, we can state our corresponding lemma, for $\rho$.

**Lemma D.6.** *For $j$ iterations of the inner loop of Algorithm 1 and stepsize $\gamma < 1/L$, we have that Algorithm 1 handles $\rho$, for $\rho$:*

$$\rho \geq -\gamma \frac{1 - (\gamma L)^j}{1 + (\gamma L)^{j+1}} \tag{33}$$

*Proof.* We start our proof from Lemma D.5, from which we get the inequality:

$$\sigma \geq \frac{(\gamma L)^j + (\gamma L)^{j+1}}{1 + (\gamma L)^{j+1}}$$

We know that for $\rho$ it holds that: $\rho \geq -\gamma + \gamma \cdot \sigma$, substituting the inequality for $\sigma$ completes the proof. $\qquad\square$

### D.1.5 Bounding $\rho$ by the number of inner steps

In this section we will convert the lower bound on $\rho$ that is presented as a function of $\gamma$ and $L$ in Lemma D.6 to a function of the inner iterations $n$

**Lemma D.7.** *For $n$ inner loop iterations and inner stepsize $\gamma = (1 - \ln n/n)/L$, we get that the minimum value for $\rho$, supported by our algorithm is at least:*

$$-\frac{1}{L}\left(1 - \frac{\ln n}{n}\right) \frac{n - 1}{n + 1 - \frac{\ln n}{n}} \tag{34}$$

*Proof.* Let us restate Lemma D.6 for the sake of exposition.

$$\rho \geq -\gamma \frac{1 - (\gamma L)^n}{1 + (\gamma L)^{n+1}} \tag{35}$$

By selecting $\gamma = x/L$, we have:

$$\rho \quad \geq \quad -\frac{1}{L} \max_{x \in (0,1]} \{x \frac{1 - x^n}{1 + x^{n+1}}\}$$

Now we will lower bound the max on the right hand side.

$$
\begin{aligned}
\max_{x \in (0,1]} \{x \frac{1 - x^n}{1 + x^{n+1}}\} \quad &\geq \quad (1 - \frac{\ln n}{n}) \frac{1 - (1 - \frac{\ln n}{n})^n}{1 + (1 - \frac{\ln n}{n})^{n+1}} \\
&\geq \quad (1 - \frac{\ln n}{n}) \frac{1 - \frac{1}{n}}{1 + \frac{1}{n}(1 - \frac{\ln n}{n})} \\
&\geq \quad (1 - \frac{\ln n}{n}) \frac{n - 1}{n + 1 - \frac{\ln n}{n}}
\end{aligned}
$$

Where in the first line we just choose to set $x = 1 - \ln(n)/n$ $\qquad \square$

### D.1.6 INFERRING ORACLE COMPLEXITY

**Lemma D.8.** *Suppose Assumption 3.1 holds. Then Algorithm 1 needs to do*

$$n = \frac{6}{1 + \rho \cdot L} \cdot \ln(\frac{3}{1 + \rho \cdot L})$$

*inner loop iterations.*

*Proof.* We start from the inequality we gave for $\rho$, Lemma D.7, which states that with $n$ inner loop iterations we can handle $\rho$, when:

$$
\begin{aligned}
\rho \quad &\geq \quad -\frac{1}{L}(1 - \frac{\ln n}{n}) \frac{n - 1}{n + 1 - \frac{\ln n}{n}} \\
&\geq \quad -\frac{1}{L}(1 - \frac{\ln n}{n}) \frac{n - 1}{n + 1}
\end{aligned}
$$

By multiplying both sides with $-L$, we get:

$$-\rho \cdot L \quad \leq \quad (1 - \frac{\ln n}{n}) \frac{n - 1}{n + 1} \tag{36}$$

Decreasing the right hand side of 36, with respect to $n$, yields worse requirements complexity wise, for $n$ and $x$. Given this observation, now let us set $n = 2 \cdot x \cdot e^x$, for some $x$, in this case, we have that:

$$\ln n = \ln(2 \cdot x \cdot e^x) = x + \ln(2 \cdot x) \leq 2x - 1 + \ln 2 \leq 2x$$

Since $2x$ is the upperbound on $\ln n$, we have that: $1 - \frac{\ln n}{n} \geq 1 - \frac{2x}{2xe^x}$. Also note that since $2xe^x \geq e^x$, we have that $\frac{2xe^x - 1}{2xe^x + 1} \geq \frac{e^x - 1}{e^x + 1}$. So:

$$(1 - \frac{\ln n}{n}) \cdot \frac{n - 1}{n + 1} \geq (1 - \frac{2x}{2xe^x}) \cdot \frac{e^x - 1}{e^x + 1}$$

So we can upperbound the number of iterations $n$ required for a certain $\rho, L$, by calculating a solution to the following equation:

$$-\rho \cdot L = (1 - \frac{1}{e^x}) \cdot \frac{e^x - 1}{e^x + 1}$$

We solve the above equation for $e^x > 1$, since we require $n > 0$. By solving for $e^x$, we get:

$$e^x = \frac{2 - \rho \cdot L + \sqrt{-\rho \cdot L}\sqrt{8 - \rho \cdot L}}{2(1 + \rho \cdot L)}$$

We can simplify the above expression by taking the upperbound for the value of $-\rho \cdot L$ which is 1. So we get:

$$e^x = \frac{3}{1 + \rho \cdot L}$$

By solving for $x$, we get:

$$x = \ln\left(\frac{3}{1 + \rho \cdot L}\right)$$

So we get that for

$$n = \frac{6}{1 + \rho \cdot L} \cdot \ln\left(\frac{3}{1 + \rho \cdot L}\right)$$

$\square$

## E    PROOFS FOR SECTION 7 (LINEAR CONVERGENCE)

**Theorem 7.3.** *Suppose Assumptions 3.1(iii) and 7.1 hold. Consider the sequence $(z^k)_{k \in \mathbb{N}}$ generated by (8) with $\lambda \in (0, 2)$, $\delta \le \rho$, $\sigma \in [0, 1 + \frac{\delta}{\gamma})$. Then, for all $z^\star \in \mathcal{Z}^\star$*

$$\text{dist}^2(z^K, \text{zer}\, S) \le (1 - \zeta)^K \text{dist}^2(z^0, \text{zer}\, S)$$

*with $\zeta = \lambda(2 - \lambda)\left(\frac{1 - \sigma + \delta/\gamma}{1 - \sigma + 1/\tau\gamma}\right)^2$.*

*Proof.* The error condition in (8) can equivalently be written as

$$\langle z^k - \bar{z}^k, \bar{v}^k \rangle \ge (1 - \sigma)\|\bar{v}^k\|^2$$

Using $\|\varepsilon^k\|^2 = \|z^k - \bar{z}^k\|^2 + \|\bar{v}^k\|^2 - 2\langle z^k - \bar{z}^k, \bar{v}^k \rangle$, due to $\varepsilon^k = z^k - \bar{z}^k - \bar{v}^k$, we have

$$\|z^k - \bar{z}^k\|^2 + \|\bar{v}^k\|^2 - \|\varepsilon^k\|^2 \ge 2(1 - \sigma)\|\bar{v}^k\|^2$$

It follows that

$$(1 - 2\sigma + \tfrac{1}{1+e})\|\bar{v}^k\|^2 \le (1 + \tfrac{1}{e})\|z^k - \bar{z}^k\|^2 \tag{37}$$

for any $e > 0$ where we have used Young's inequality through the inequality

$$-\|a - b\|^2 \le -\tfrac{1}{1+e}\|a\|^2 + \tfrac{1}{e}\|b\|^2$$

which holds for any $a, b \in \mathbb{R}^d$ and $e > 0$. By optimizing $a = \frac{\sigma}{1-\sigma}$ in (37) and rearranging we obtain

$$\|z^k - \bar{z}^k\| \ge \xi\|\bar{v}^k\| \tag{38}$$

with $\xi = 1 - \sigma$.

The update (8) is constructed as the KM iteration on the projection operator $P$ in (7). Thus, from the Fejér monotonicity of the KM iteration established in (27) we have

$$\|z^{k+1} - z^\star\|^2 \le \|z^k - z^\star\|^2 - \lambda_k(2 - \lambda_k)\|Pz^k - z^k\|^2$$
$$= \|z^k - z^\star\|^2 - \lambda_k(2 - \lambda_k)\alpha_k^2\|\bar{v}^k\|^2$$

from which it follows that

$$\text{dist}^2(z^{k+1}, \text{zer}\, S) \le \text{dist}^2(z^k, \text{zer}\, S) - \lambda_k(2 - \lambda_k)\alpha_k^2\|\bar{v}^k\|^2. \tag{39}$$

Using the triangle inequality

$$\text{dist}(z^k, \text{zer}\, S) \le \|z^k - z^\star\| \le \|\bar{z}^k - z^\star\| + \|z^k - \bar{z}^k\|$$
$$\overset{(38)}{\le} \|\bar{z}^k - z^\star\| + \xi\|\bar{v}^k\|$$
$$(\text{Assumption } 7.1) \le (\tfrac{1}{\tau\gamma} + \xi)\|\bar{v}^k\| \tag{40}$$

with $z^\star = \mathbf{\Pi}_{\text{zer}\, S}(\bar{z}^k)$. Combining (40) and (39) we obtain

$$\text{dist}^2(z^{k+1}, \text{zer}\, S) \le (1 - \zeta_k)\text{dist}^2(z^k, \text{zer}\, S)$$

with $\zeta_k = \lambda_k(2 - \lambda_k)\alpha_k^2 \frac{1}{(1/\tau\gamma + \xi)^2} \ge \lambda_k(2 - \lambda_k)(1 + \frac{\delta}{\gamma} - \sigma)^2 \frac{1}{(1/\tau\gamma + \xi)^2}$ which uses $\alpha_k \ge 1 + \frac{\delta}{\gamma} - \sigma$ from Proposition 5.1. Taking $\lambda_k = \lambda$ completes the proof. $\square$

# F   PROOFS FOR SECTION 8 (TIGHTNESS)

To prove the lower bound we rely on the following operator based on Pethick et al. (2022, Ex. 5).

**Example F.1.** *Consider the following linear operator:*

$$Fz = (ay + bx, by - ax),\tag{41}$$

*where $z = (x, y)$, $b < 0$ and $a > 0$. It is straightforward to verify that*

(i) *Assumption 3.1(i) is satisfied with $L = \sqrt{a^2 + b^2}$.*

(ii) *Assumption 3.1(iii) is satisfied with $\rho = \frac{b}{a^2+b^2}$.*

**Theorem 8.1.** *Consider a sequence $(z^k)_{k\in\mathbb{N}}$ generated according to rPPA with $\gamma > 0$ and $\bar{\alpha}_k > 0$ for all $k \in \mathbb{N}$. Let $\rho \leq -(1 - \frac{\bar{\alpha}_k}{2})\gamma$. Then, there exists an operator $S : \mathbb{R}^d \to \mathbb{R}^d$ for any $d > 1$ satisfying Assumption 3.1(iii) for which the sequence will not converge.*

*Proof.* We follow the argument in Pethick et al. (2022, Thm. 3.4) closely. The update rule rPPA is a linear operator by linearity of the operator $F$. Specifically,

$$\begin{aligned}
T : &= (1 - \bar{\alpha}_k)I + \bar{\alpha}_k(I + \gamma F)^{-1} \\
&= \begin{pmatrix} \frac{\bar{\alpha}_k(b\gamma+1)-(\bar{\alpha}_k-1)(a^2\gamma^2+b^2\gamma^2+2b\gamma+1)}{a^2\gamma^2+b^2\gamma^2+2b\gamma+1} & -\frac{a\bar{\alpha}_k\gamma}{a^2\gamma^2+b^2\gamma^2+2b\gamma+1} \\ \frac{a\bar{\alpha}_k\gamma}{a^2\gamma^2+b^2\gamma^2+2b\gamma+1} & \frac{\bar{\alpha}_k(b\gamma+1)}{a^2\gamma^2+b^2\gamma^2+2b\gamma+1} - \bar{\alpha}_k + 1 \end{pmatrix}
\end{aligned}\tag{42}$$

A linear dynamical system is globally asymptotically stable iff the spectral radius of the linear mapping is strictly less than 1. Let $\lambda_1, \lambda_2$ be the eigenvalues of $T$. Then the spectral radius is the largest absolute value of the eigenvalues. For $T$ we have,

$$\begin{aligned}
\max_{i\in\{1,2\}} |\lambda_i| &= \frac{\sqrt{(a^2\gamma^2 + (b\gamma + 1)^2)((\bar{\alpha}_k - 1)\gamma((\bar{\alpha}_k - 1)\gamma(a^2 + b^2) - 2b) + 1)}}{\gamma^2(a^2 + b^2) + 2b\gamma + 1} \\
&= \sqrt{\frac{(\bar{\alpha}_k - 1)\gamma L^2((\bar{\alpha}_k - 1)\gamma - 2\rho) + 1}{\gamma L^2(\gamma + 2\rho) + 1}},
\end{aligned}\tag{43}$$

where we have used that $a = \sqrt{L^2 - L^4\rho^2}$ and $b = L^2\rho$. Solving for $\max_i |\lambda_i| < 1$, we obtain the requirement $\rho < -(1 - \frac{\bar{\alpha}_k}{2})\gamma$ for convergence of the discrete dynamical system. On the other hand, since (42) is a linear system, we simultaneously learn that picking $\gamma$ any smaller would imply non-convergence through $\max_i |\lambda_i| \geq 1$ (given $z^0 \neq 0$). Noting that problem (42) can be embedded into a higher dimension completes the proof. We provide an accompanying Mathematica notebook for verifying the above steps. $\qquad\square$

# G   RELAXING $\rho$ THROUGH A COORDINATE-WISE UPDATE

In this section we show that our implicit scheme (8) can also be used to capture a similar construction to the primal dual extragradient algorithm in Pethick et al. (2023a). The coordinatewise update we will develop is important for relaxing the Lipschitz condition, which in term can relax the condition on the weak MVI parameter $\rho$.

To allow for coordinate specific stepsizes we will introduce the positive definite stepsize matrix $\Gamma \in \mathbb{R}^{d\times d}$. Note that this generalization of (8) only affects the constants in the resulting convergence guarantee, but is not necessary for relaxing the condition on $\rho$. In other words, (8) can directly capture the Gauss-Seidel type method developed in this section for a particular choice of the error $\varepsilon^k = \gamma(M_{z^k}(z^k) - M_{z^k}(\bar{z}^k))$ where $M$ is defined in (48). For simplicity we will consider a nonadaptive variant with some $\alpha > 0$:

$$\begin{aligned}
\text{find} \quad & \bar{z}^k \in \mathbb{R}^d \quad \text{and} \quad \bar{v}^k \in S\bar{z}^k \\
\text{s.t.} \quad & \Gamma^{-1}(z^k - \bar{z}^k) = \bar{v}^k + \varepsilon^k \quad \text{and} \quad -\langle\varepsilon^k, \bar{v}^k\rangle_\Gamma \leq \sigma\|\bar{v}^k\|_\Gamma^2 \\
\text{compute} \quad & z^{k+1} = z^k - \alpha\Gamma\bar{v}^k
\end{aligned}\tag{44}$$

**Theorem G.1.** *Suppose Assumptions 3.1(ii) and 3.1(iii) hold. Consider the sequence $(z^k)_{k\in\mathbb{N}}$ generated by (44) with $\alpha > 0$ and $\rho > -\bar{\gamma}(1 - \sigma - \frac{\alpha}{2})$ where $\bar{\gamma}$ is the smallest eigenvalue of $\Gamma$. Then, for all $z^\star \in \mathcal{Z}^\star$*

$$\min_{k\in\{0,\dots,K-1\}} \text{dist}_\Gamma(0, S\bar{z}^k)^2 \leq \frac{\|z^0 - z^\star\|_{\Gamma^{-1}}^2}{2\alpha(1 - \sigma - \frac{\alpha}{2} + \frac{\rho}{\bar{\gamma}})K}. \tag{45}$$

*Remark G.2.* Note that well-definedness and single-valuedness of the proximal update still needs to be ensured.

*Proof.* Recall that $\bar{v}^k \in S\bar{z}^k$ and the error condition

$$-\langle \varepsilon^k, \bar{v}^k \rangle_\Gamma \leq \sigma \|\bar{v}^k\|_\Gamma^2 \tag{46}$$

The update in (44) yields

$$\begin{aligned}
\|z^{k+1} - z^\star\|_{\Gamma^{-1}}^2 &= \|z^k - z^\star\|_{\Gamma^{-1}}^2 + \alpha^2\|\bar{v}^k\|_\Gamma^2 - 2\alpha\langle \bar{v}^k, z^k - z^\star\rangle \\
&= \|z^k - z^\star\|_{\Gamma^{-1}}^2 + \alpha^2\|\bar{v}^k\|_\Gamma^2 - 2\alpha\langle \bar{v}^k, z^k - \bar{z}^k\rangle - 2\alpha\langle \bar{v}^k, \bar{z}^k - z^\star\rangle \\
&= \|z^k - z^\star\|_{\Gamma^{-1}}^2 - 2\alpha(1 - \tfrac{\alpha}{2})\|\bar{v}^k\|_\Gamma^2 - 2\alpha\langle \bar{v}^k, \varepsilon^k\rangle_\Gamma - 2\alpha\langle \bar{v}^k, \bar{z}^k - z^\star\rangle \\
\overset{(46)}{\leq} &\|z^k - z^\star\|_{\Gamma^{-1}}^2 - 2\alpha(1 - \sigma - \tfrac{\alpha}{2})\|\bar{v}^k\|_\Gamma^2 - 2\alpha\langle \bar{v}^k, \bar{z}^k - z^\star\rangle \\
\overset{\text{(Assumption 3.1(iii))}}{\leq} &\|z^k - z^\star\|_{\Gamma^{-1}}^2 - 2\alpha(1 - \sigma - \tfrac{\alpha}{2})\|\bar{v}^k\|_\Gamma^2 - 2\alpha\rho\|\bar{v}^k\|^2 \\
\leq &\|z^k - z^\star\|_{\Gamma^{-1}}^2 - 2\alpha(1 - \sigma - \tfrac{\alpha}{2} + \tfrac{\rho}{\bar{\gamma}})\|\bar{v}^k\|_\Gamma^2
\end{aligned} \tag{47}$$

where the last inequality uses that $\|\bar{v}^k\|^2 \leq \frac{1}{\bar{\gamma}}\|\bar{v}^k\|_\Gamma^2$ with $\bar{\gamma}$ being the smallest eigenvalue of $\Gamma$. Rearranging, summing and telescoping completes the proof. □

Instead of making Lipschitz continuity assumptions on $F$ in $S = F + A$ directly, we consider the following operator,

$$M_u(z) := F(z) - Q_u(z), \tag{48}$$

where the operator $Q_u : \mathbb{R}^d \to \mathbb{R}^d$ is to be defined.

**Assumption G.3.** *The operator $M$ as defined in (48) is $L_M$-Lipschitz with $L_M \leq 1$ with respect to a positive definite matrix $\Gamma \in \mathbb{R}^{d\times d}$, i.e.*

$$\|M_z(z) - M_z(z')\|_\Gamma \leq L_M\|z - z'\|_{\Gamma^{-1}} \quad \forall z, z' \in \mathbb{R}^d. \tag{49}$$

We will write the update in terms of $H_u(z) := \Gamma^{-1}z - M_u(z)$, which will be important in order to relax the Lipschitz conditions. With $M_u$ defined, it is straightforward to establish that $H_u$ is $1/2$-cocoercive and strongly monotone as long as $M_u$ is $L_M$-Lipschitz continuous as done in (Pethick et al., 2023a) (see Lemma G.4). This provides a strict generalization of Lemma A.4 concerning the forward operator $H := \text{id} - \gamma F$.

**Lemma G.4** (Pethick et al. (2023a, Lem. F.2))**.** *Suppose Assumption G.3 holds and let $\Gamma \in \mathbb{R}^{d\times d}$ be a positive definite matrix. Then, the mapping $H_u(z) := \Gamma^{-1}z - M_u(z)$ is $1/2$-cocoercive, i.e.,*

$$\langle H_z(z') - H_z(z), z' - z\rangle \geq \tfrac{1}{2}\|H_z(z') - H_z(z)\|_\Gamma^2 \quad \forall z, z' \in \mathbb{R}^d. \tag{50}$$

*Proof.* By expanding using (48),

$$H_z(z) - H_z(z') = \Gamma^{-1}(z - z') - (M_z(z) - M_z(z')). \tag{51}$$

Using this we can show cocoercivity,

$$\begin{aligned}
\langle H_z(z') - H_z(z), z' - z\rangle &= \langle H_z(z') - H_z(z), H_z(z') - H_z(z) - (M_z(z) - M_z(z'))\rangle_\Gamma \\
\overset{(51)}{=} &\tfrac{1}{2}\|H_z(z') - H_z(z)\|_\Gamma^2 + \tfrac{1}{2}\|z' - z\|_{\Gamma^{-1}}^2 - \tfrac{1}{2}\|M_u(z) - M_u(z')\|_\Gamma^2 \\
\overset{\text{(Assumption G.3)}}{\geq} &\tfrac{1}{2}\|H_z(z') - H_z(z)\|_\Gamma^2 + \tfrac{1}{2}(1 - L_M^2)\|z' - z\|_{\Gamma^{-1}}^2
\end{aligned} \tag{52}$$

This establishes the claim. □

Specifically, pick the error $\varepsilon^k = M_{z^k}(z^k) - M_{z^k}(\bar{z}^k)$ in (44) such that the update rule defining $\bar{z}^k$ reduces to a preconditioned update asking for a $\bar{z}^k \in \mathbb{R}^d$ such that

$$H_{z^k}(z^k) - H_{z^k}(\bar{z}^k) = \bar{v}^k \in S\bar{z}^k \tag{53}$$

Thus, the update (44) for the particular choice of $\varepsilon^k$ reduces to the following with $\Gamma \in \mathbb{R}^{d \times d}$ and $\alpha > 0$:

$$
\begin{aligned}
\text{find} \quad & \bar{z}^k \in \mathbb{R}^d, v^k \in S\bar{z}^k \\
\text{s.t.} \quad & H_{z^k}(z^k) - H_{z^k}(\bar{z}^k) = \bar{v}^k \\
\text{compute} \quad & \bar{z}^{k+1} = z^k - \alpha\Gamma(H_{z^k}(z^k) - H_{z^k}(\bar{z}^k))
\end{aligned}
\tag{54}
$$

We obtain the following corollary by specializing Theorem G.1 to the update in (54).

**Corollary G.5.** *Suppose Assumptions 3.1(ii), 3.1(iii) and G.3 hold. Consider the sequence $(z^k)_{k \in \mathbb{N}}$ generated by (54), $\alpha > 0$ and $\rho > -\frac{(1-\alpha)\bar{\gamma}}{2}$ where $\bar{\gamma}$ is the smallest eigenvalue of $\Gamma$. Then, for all $z^\star \in \mathcal{Z}^\star$*

$$\min_{k \in \{0,\dots,K-1\}} \text{dist}_\Gamma(0, S\bar{z}^k)^2 \leq \frac{\|z^0 - z^\star\|_{\Gamma^{-1}}^2}{\alpha(1 - \alpha + \frac{2\rho}{\bar{\gamma}})K}. \tag{55}$$

*Proof.* Let us compute the error tolerance $\sigma$ for the particular choice of the error $\varepsilon^k$. From $1/2$-cocoercive due to Lemma G.4 we immediately have

$$
\begin{aligned}
\tfrac{1}{2}\|H_{z^k}(z^k) - H_{z^k}(\bar{z}^k)\|_\Gamma^2 &\leq \langle H_{z^k}(z^k) - H_{z^k}(\bar{z}^k), z^k - \bar{z}^k \rangle \\
&= \langle H_{z^k}(z^k) - H_{z^k}(\bar{z}^k), M_{z^k}(z^k) - M_{z^k}(\bar{z}^k) \rangle_\Gamma \\
&\quad + \|H_{z^k}(z^k) - H_{z^k}(\bar{z}^k)\|_\Gamma^2
\end{aligned}
\tag{56}
$$

which is equivalent to $-\langle \bar{v}^k, \varepsilon^k \rangle_\Gamma \leq \sigma\|\bar{v}^k\|_\Gamma^2$ with $\sigma = \frac{1}{2}$. We obtain the result by specializing Theorem G.1 for the particular $\sigma$. $\square$

Observe that the weak MVI parameter $\rho$ in Corollary G.5 depends on the Lipschitz constant of operator $M$ rather than Lipschitz operator $F$ directly, through the smallest eigenvalue $\bar{\gamma}$ of the stepsize matrix $\Gamma$. This is important because $\bar{\gamma}$ can potentially be larger than $\bar{\gamma} = \frac{1}{L}$, which is otherwise required if we were relying on $L$-Lipschitz of $F$.

In the following, we will choose an asymmetric $Q$ in $M$, that leads to a well-defined (Gauss-Seidel) update for $\bar{z}^k$, and then subsequently show the implication for the Lipschitz condition. For simplicity, we first illustrate the idea for minimax problem resulting in a block-coordinate update with only two blocks.

**Minimax** Consider the following minimax problem

$$\min_{x \in \mathcal{X}} \max_{y \in \mathcal{Y}} \phi(x, y). \tag{57}$$

We will pick an asymmetric $Q$,

$$Q_z(\bar{z}) = (0, -\nabla_y \phi(\bar{x}, y)) \tag{58}$$

with $z = (x, y)$ and $\bar{z} = (\bar{x}, \bar{y})$, in which case (53) is well-defined and becomes an alternating update

$$\bar{z}^k = (H_{z^k} + S)^{-1} H_{z^k}(z^k) \quad \Leftrightarrow \quad \begin{cases} \bar{x}^k = \mathbf{\Pi}_\mathcal{X}(x^k - \Gamma_1 \nabla_x \phi(x^k, y^k)) \\ \bar{y}^k = \mathbf{\Pi}_\mathcal{Y}(y^k + \Gamma_2 \nabla_y \phi(\bar{x}^k, y^k)) \end{cases} \tag{59}$$

where we have rescaled the iterates for convenience by defining $\Gamma^{-1}\bar{z}^k = (\bar{x}^k, \bar{y}^k)$. We furthermore have that the desired Lipschitz assumption on the operator $M$ (Assumption G.3) holds provided

$$
\begin{aligned}
\|\nabla_x \phi(x, y) - \nabla_x \phi(x', y')\|^2 &\leq L_{xx}^2 \|x - x'\|^2 + L_{xy}^2 \|y - y'\|^2 \\
\|\nabla_y \phi(x, y) - \nabla_y \phi(x, y')\|^2 &\leq L_{yy}^2 \|y - y'\|^2
\end{aligned}
$$

Specifically, we have that $L_M = \sqrt{\max\{L_{xx}^2\|\Gamma_1\|^2, L_{xy}^2\|\Gamma_2\|^2 + L_{yy}^2\|\Gamma_2\|^2\}}$ which can be smaller than the Lipschitz constant $L$ of the operator $F$.

**Coordinatewise**   More generally, we can consider a block-coordinatewise update for a general $F(z) = (F_1(z), \ldots, F_m(z))$. Consider the decomposition $z = (z_1, \ldots, z_m) \in \mathbb{R}^d$ and define the shorthand notation $z_{\leq i} := (z_1, z_2, \ldots, z_m)$ and $z_{\geq i} := (z_i, \ldots, z_m)$ for the truncated vector. Moreover suppose that $A$ decomposes into $Az = (A_1 z_1, \ldots, A_m z_m)$ with $A_i : \mathbb{R}^{d_i} \rightrightarrows \mathbb{R}^{d_i}$ maximally monotone and define $\Gamma = \mathrm{blkdiag}(\Gamma_1, \ldots, \Gamma_m)$ where $\Gamma_i \in \mathbb{R}^{d_i \times d_i}$ are positive definite matrices. We will pick an asymmetric $Q$,

$$Q_z(\bar{z}) = (0, F_1(\bar{z}_1, z_{\geq 2}), F_2(\bar{z}_1, \bar{z}_2, z_{\geq 3}), \ldots, F_m(\bar{z}_{\leq m-1}, z_m)) \tag{60}$$

in which case (53) is well-defined (see Pethick et al. (2023a, Lem. F.1)) and becomes a Gauss-Seidel update

$$\bar{z}^k = (H_{z^k} + S)^{-1} H_{z^k}(z^k) \; \Leftrightarrow \; \bar{z}_i^k = \begin{cases} (\Gamma_1^{-1} + A_1)^{-1}(z_1^k - F_1(z^k)) & \text{if } i = 1 \\ (\Gamma_i^{-1} + A_i)^{-1}(z_i^k - F_{i-1}(\bar{z}_{\leq i-1}^k, z_{\geq i}^k)) & \text{if } i = 2, \ldots, m \end{cases} \tag{61}$$

We furthermore have that the desired Lipschitz continuous assumption on the operator $M$ (Assumption G.3) holds provided

$$\|F_i(z) - F_i(z_{\leq i-1}, z'_{\geq i})\|^2 \leq \sum_{j=i}^m L_{z_i z_j}^2 \|z_j - z'_j\|^2 \quad \forall i \in [m]$$

Specifically, we have that $L_M^2 = \max_{i \in [m]} \{\sum_{j \in [i]} L_{z_j z_i}^2 \|\Gamma_i\|^2\}$ which can be smaller than the Lipschitz constant $L$ of the operator $F$.

