# OpenReview forum: "Efficient Interpolation between Extragradient and Proximal Methods for Weak MVIs"
_ICLR.cc/2025/Conference — ICLR 2025 Poster_

### Official Review · Reviewer_F8GR · 2024-10-15

**Soundness:** 2
**Presentation:** 2
**Contribution:** 3
**Rating:** 6
**Confidence:** 3

**Summary:**

This paper studies the nonmonotone games satisfying the weak Minty variational inequality (MVI). The authors propose an inexact hybrid proximal extragradient method. The method converges with the rate of $O(1/\epsilon)$ in the maximal range of $\rho$. Compared with Alacaoglu et al. (2024), the method in this work removes a $O(\log(1/\epsilon))$ factor. The authors also show that linear convergence is automatically achieved under their proposed inexact conditions.

**Strengths:**

The contribution and improvement over previous methods are clear. The analysis in this paper also seems interesting to me.

If I understand correctly, using exact proximal point methods to extend the range of $\rho$ to $(-1/L,\infty)$ seems straightforward.
An additional $O(1/\epsilon)$ naturally appears if using inexact proximal point methods. Therefore the result in Alacaoglu et al. (2024) seems not very surprising.
But the result without such a log factor seems much more challenging and interesting to me.

**Weaknesses:**

1. This paper only considers the deterministic case, which seems to be easier than the stochastic case considered by some other works such as [1,2].

2.  Alacaoglu et al. (2024) also consider the cohypomonotone case, with an accelerated rate. The acceleration seems more interesting and challenging.

[1] Alacaoglu, Ahmet, Donghwan Kim, and Stephen J. Wright. "Extending the Reach of First-Order Algorithms for Nonconvex Min-Max Problems with Cohypomonotonicity." arXiv preprint arXiv:2402.05071 (2024).

[2] Chen, Lesi, and Luo Luo. "Near-optimal algorithms for making the gradient small in stochastic minimax optimization." arXiv preprint arXiv:2208.05925 (2022).

**Questions:**

Most of my questions are about related works.
1. Have the authors considered the stochastic case like [1,2]. Is it possible to remove the log factor using your framework?
2. It seems that a similar idea (using proximal methods) also appears in [2]. Although the result in [2] writes the assumption $\rho > - \frac{1}{2L}$, it seems their proposed algorithm can also be used in the range $\rho > - \frac{1}{L}$ (due to the use the proximal methods) just by changing the constants in their algorithm. Also, [1] applies very a similar MLMC technique as used in [2]. Could the author please tell me if my understanding is correct and compare these related works for me?
3. The method in this paper looks similar to [3]. What are the differences between the method in this paper and [3]?

[1] Alacaoglu, Ahmet, Donghwan Kim, and Stephen J. Wright. "Extending the Reach of First-Order Algorithms for Nonconvex Min-Max Problems with Cohypomonotonicity." arXiv preprint arXiv:2402.05071 (2024).

[2] Chen, Lesi, and Luo Luo. "Near-optimal algorithms for making the gradient small in stochastic minimax optimization." arXiv preprint arXiv:2208.05925 (2022).

[3] Monteiro, Renato DC, and Benar F. Svaiter. "Iteration-complexity of a Newton proximal extragradient method for monotone variational inequalities and inclusion problems." SIAM Journal on Optimization 22.3 (2012): 914-935.

---

> ### Author Response · Authors · 2024-11-20
>
> We thank the reviewer for the feedback and address all remaining concerns below.
>
> > Extensions to the stochastic case and cohypomonotone case
>
> These are very promising directions, but they each require their own treatment.
> In general our work opens up for a range of extensions:
>
> - the special case of cohypomonotonicity
> - the stochastic case
> - higher order methods (see the last question)
>
> Let us comment on the stochastic and cohypomontone case specifically.
>
> _Stochastic_ Even for $\rho > -1/(2L)$, the stochastic case is still not closed. The main two approaches currently are:
>
> - Use increasing batch size (e.g. [1], [2], Diakonikolas et al. 2021, Böhm 2022)
> - Use fixed batch size but make additional smoothness assumptions (Pethick et al. 2023a)
>
> Avoiding increasing batch size is generally preferred, but it is currently unknown whether fixed batch size is possible without additional assumptions.
> It is thus not entirely clear which approach should be taken in the stochastic case.
>
> _Cohypomonotone_ The cohypomonotone case is definitely interesting (and we indeed mention it as a promising direction in the conclusion), but we politely disagree that it is _more_ interesting, since the accelerated methods comes at a price:
>
> - The cohypomonotone case rules out e.g. the motivating example in Diakonikolas et al. 2021.
> - The accelerated methods do not enjoy the same adaptivity to error bound conditions (see Section 7)
>
> We would rather say that both weak MVI methods and methods specialized for cohypomonotone problems are independently interesting to study. We have focused on methods for weak MVI as this already proved challenging.
> Hopefully, our work can inspire developments for the special case of cohypomonotone problems as well.
>
> > 1. This paper only considers the deterministic case, which seems to be easier than the stochastic case considered by some other works such as [1,2].
> > 2.  Alacaoglu et al. (2024) also consider the cohypomonotone case, with an accelerated rate. The acceleration seems more interesting and challenging.
>
>
> **Questions**:
>
> > 1. "Have the authors considered the stochastic case like [1,2]. Is it possible to remove the log factor using your framework?"
>
> Removing the logarithmic factor in the stochastic case by extending this work might indeed be possible and it is a very interesting direction. However, note the comment above, that the stochastic case is somewhat open even for $\rho > -1/(2L)$.
>
> > 2. Comparison with [2] and its connection with [1]
>
> The method in [2] indeed uses a proximal update, but is restricted to $\rho > -1/(2L)$, since no relaxation is used. In fact, our Theorem 8.1 provides a lower bound even in the deterministic setting (see Remark 8.2). Also note that the method does not include an extragradient computation, which is crucial for removing the logarithmic factor in the complexity (please see Eq. 9-10 for comparison with inexact proximal based methods).
>
> Regarding the stochastic case, [2] indeed uses MLMC similarly to [1]. There are some subtle differences regarding parameter choices (see e.g. the comment after Estimator 4.4 in [1]).
>
> > 3. "The method in this paper looks similar to [3]. What are the differences between the method in this paper and [3]?"
>
> The approach [3] can be seen as a special case of our implicit scheme in Eq. 8 in the following sense:
>
> - [3] is a special case of Solodov & Svaiter's HPE method (which applies to the monotone case). In [3] they specifically considers when the proximal update is approximated with a second order update.
> - Recall, that our method is a generalization of Solodov & Svaiter HPE method by extending it to weak MVIs and relaxing the error condition (See e.g. our Remark 4.1 and the new Appendix B, which now contains the special cases explicitly).
>
> Our implicit scheme Eq. 8 opens up for similarly considering higher order methods for weak MVIs.

---

> > ### Comment · Reviewer_F8GR · 2024-11-22
> > **Thanks for the authors' rebuttal**
> >
> > I thank the authors'  response. But my first question remains unaddressed.
> >
> > From [1] (Appendix A.1), "It is enough to show that $F$ is monotone if and only if $F_{\eta}$ is $-\frac{1}{\eta}$ comonotone." So one can take any $\eta < L$ and solve the problem that $\rho > -1/L$. I think it is unfair to say "the method in [2] indeed uses a proximal update, but is restricted to $\rho > -1/(2L)$, since no relaxation is used." From my point of view, one can simply change the regularization parameter from $2L$ to any constant that is strictly smaller than $L$.
> >
> > As also pointed out by Reviewer kMEF, extending the range from $\rho \ge -1/(2L)$ to $\rho > - \frac{1}{L}$ may not be a significant problem itself.
> > To me, it is very strange that a line of works in 2023-2024 tries to address a problem that is not that important and seems to be trivial in 2021-2022 [1-2].
> >
> > [1] Lee S, Kim D. Fast extra gradient methods for smooth structured nonconvex-nonconcave minimax problems. In NeurIPS， 2021.
> >
> > [2]  Chen, Lesi, and Luo Luo. "Near-optimal algorithms for making the gradient small in stochastic minimax optimization." arXiv preprint arXiv:2208.05925 (2022).

---

> > > ### Author Response · Authors · 2024-11-22
> > >
> > > We thank the reviewer for their response.
> > >
> > > >  I thank the authors' response. But my first question remains unaddressed.
> > >
> > > No, we have not yet looked into the stochastic case, since the deterministic case already required considerable work. For a stochastic extension of Algorithm 1 there are two sources of stochasticity to consider:
> > >
> > > - the inner solver for approximating the proximal operator
> > > - the extragradient computation $\bar v^k \in S\bar z^k$
> > >
> > > > From my point of view, one can simply change the regularization
> > >
> > > This is a good point – our comment and the lower bound were regarding applying the proximal operator to $F$ directly.
> > >
> > > The proposed line of argument is essentially: $F$ is $-1/\eta$-comonotone iff $F_\eta$ is monotone iff the resolvent $(\operatorname{id} + F_\eta)^{-1}$ is firmly nonexpansive from which convergence follows. It seems plausible that this would generalize to the case of weak MVI. We will make sure to cite [2] and comment on the fact that $\rho > -1/L$ can be attained using their result.
> > >
> > > > To me, it is very strange that a line of works in 2023-2024 tries to address a problem that is not that important and seems to be trivial in 2021-2022 [1-2].
> > >
> > > We will highlight that $\rho > -1/L$  is possible with techniques developed in 2021-2022 prior to the cited work from 2023-2024.
> > > It is maybe worth stating, that this observation does not reduce any of our contributions or claims, since our paper is concerned with removing the logarithmic factor, which still seems nontrivial.

---

> > > > ### Comment · Reviewer_F8GR · 2024-11-23
> > > > **Thanks for you prompt response**
> > > >
> > > > I acknowledge the paper's contribution of removing the logarithmic factor. I decide to raise the score.

---

### Official Review · Reviewer_bh4m · 2024-10-29

**Soundness:** 4
**Presentation:** 3
**Contribution:** 3
**Rating:** 6
**Confidence:** 4

**Summary:**

Non monotone games, which are operators that are non convex and non concave appear in several applications and are proven to be hard to solve. One important class of problems within this is the class of problems satisfying the weak Minty Variational Inequality. Thiis was one of the counter examples proving the hardness of the broader family of problems. The weak minty condition is a relaxation of monotonicity, i.e., roughly for some $\rho$, we require for all solutions $z^{\star}$ of an operator S,
$$ \langle S(z) , z-z^{\star}\rangle \geq \rho \|S(z)\|_2^2.$$

There has been a flurry of work that has tried to give algorithms for different values of $\rho$. When the operator is $L$-Lipschitz, the relaxed extragradient method allows $rho > -1/2L$ and the relaxed proximal point algorithm allows a larger range, $\rho > -1/L$ but loses an additional $log 1/\epsilon$ factor. This paper considers the range $-1/L<\rho <-1/2L$ and gives an algorithm that gets rid of this extra $log 1/\epsilon$ factor. Their method also unifies the two methods mentioned above. The additionally also obtain some kind of linear convergence under additional assumptions that I dont fully follow.

Their algorithm follows an inexact proximal point algorithm, where they require the errors of every prox step to be small in order to obtain convergence. Normally this would involve increasing the number of iterations, but the authors get away with this using “half-space projections” inspired by Solodov and Svaiter’99.

**Strengths:**

The paper is able to unify a few methods to get one coherent method for solving the Minty Variational Inequality problem to an optimal rate. They also improve a $\log 1/\epsilon$ factor for an important range of parameters. It is overall reasonable well written, though I would hope that the next version has background which is easier to parse for a newcomer in the field.

**Weaknesses:**

I think the paper would improve if there is a more detailed summary of what the previous methods do. I understand the space limitations in the main text but maybe something in the appendix would help. Currently there is a brief overview in the introduction but since the method unifies certain methods, it would be nice to know in somewhat more detail what the unification is.

**Questions:**

Some suggestions to the authors:
1. Please check the introduction, it has several grammatical errors.
2. If possible please introduce the halfspace method. It seems to show up without introduction and it is not clear what it means in Section 4
3. It is also not clear to me why the halfspace method works in getting better accuarcy? Also is it the same as the Ellipsoid method? I can imagine that the ellipsoid method would work the same way for convex functions.
4. I dont follow what the linear convergence means. Is it for the inner loop?

---

> ### Author Response · Authors · 2024-11-20
>
> We thank the reviewer for the feedback and address all remaining concerns below.
>
> > Overview of unification
>
> We thank the reviewer for the suggestion.
> We have added an appendix (Appendix B) clarifying how to obtain existing algorithms such as the relaxed proximal point algorithm, the method of Solodov & Svaiter, AdaptiveEG+, forward-backward-forward, relaxed extragradient/extragradient+ from our unified framework.
> Each of the methods are presented in the notation of the paper to make the comparison clear and explicitly states the choice of parameter, approximation error, etc.
> We believe this should help contextualizing our unification.
>
> **Questions**:
>
> > Typos in introduction
>
> Fixed.
>
> > "If possible please introduce the halfspace method. It seems to show up without introduction and it is not clear what it means in Section 4."
>
> We introduce the halfspace projection in two steps:
>
> - The paragraph l. 187-192 describes the high-level idea of the projection (correcting for the inexactness of the approximate proximal update)
> - The following l. 192-208 makes the exact form of our halfspace projection precise (the use of an extragradient computation)
>
> We have reworded the paragraph l. 187-192, to make it clearer that the halfspace projection deals with the inexactness of _the proximal point update_.
>
> > "It is also not clear to me why the halfspace method works in getting better accuarcy?"
>
> The main reason for why the halfspace projection method removes the logarithmic factor in the complexity is commented on at Eq. 6 in Section 4. That is, regardless of the inexactness of the proximal update, we can project onto a set that moves the iterate closer to the solutions. As a result, the errors wont accumulate in the analysis as is otherwise the case for the relaxed inexact proximal point algorithm.
>
> > Comparison with ellipsoid method
>
> The methods are distinct, which can be seen by the fact that the halfspace projection divides the space with a hyperplane, whereas the ellipsoid method constructs an ellipsoid at every iteration.
>
> > "I dont follow what the linear convergence means. Is it for the inner loop?"
>
> Linear convergence is established for the full algorithm (Algorithm 1) through Theorem 7.3 under the error bound condition.
> This assumption e.g. captures the the weak MVI lower bound example of Pethick et al. (2022), explaining the fast convergence observed in practice.
> This automatic adjustment when favorable structure is present is something that neither the relaxed inexact proximal point algorithm or Halpern based approaches enjoy (please see Section 7 for more detail).

---

> > ### Comment · Reviewer_bh4m · 2024-11-26
> >
> > I thank the authors for their response and I have decided to maintain my score.

---

### Official Review · Reviewer_Gq7N · 2024-11-03

**Soundness:** 3
**Presentation:** 2
**Contribution:** 3
**Rating:** 6
**Confidence:** 4

**Summary:**

This paper introduces an inexact proximal point algorithm with an error correction mechanism, which guarantees convergence for minimax problems that satisfy the weak Minty variational inequality (MVI) with a parameter  $\rho>-1/L$. The proposed method achieves a convergence rate of  $\mathcal O(1/\epsilon)$ without the logarithmic complexity factor found in previous approaches, making it a promising solution for weak MVIs in constrained settings.

**Strengths:**

* The proposed algorithm achieves better iteration complexity by avoiding the logarithmic factor present in previous methods.

* The performance results shown in Figure 2 are impressive and highlight the algorithm's effectiveness.

* The scheme unifies the analysis of the relaxed extragradient method, the relaxed proximal point algorithm (rPPA), and classical methods for monotone problems.

**Weaknesses:**

* It would be helpful if the authors elaborated on the intuition behind the necessity of multiple inner loops when $\rho<-1/2L$. Further insight into why these loops are essential could enhance understanding of the algorithm’s design choices.

* In Section 8, the authors mention the possibility of extending the range of $\rho$ beyond $1/L$. Including experimental results to illustrate this potential would strengthen the paper’s claims and provide evidence for the method’s flexibility under various settings of $\rho$.

*  The paper would benefit from additional empirical evaluation to demonstrate its applicability to complex real-world problems. Alternatively, a more explicit discussion on the practical relevance of these theoretical results could help clarify their significance for challenging problem settings.

* The paper’s notation is somewhat confusing, as the authors sometimes use the range of $\gamma$ to imply constraints on
$\rho$ and other times directly specify the range of $\rho$. Since $\rho$ and $\gamma$ are related, a unified notation throughout would make the technical content easier to follow, especially given the paper’s density.

**Questions:**

See weaknesses.

---

> ### Author Response · Authors · 2024-11-20
>
> We thank the reviewer for the feedback and address all remaining concerns below.
>
> > On the necessity of multiple inner loops
>
> Section 6 is dedicated to explaining this. In short:
>
> - The goal is to satisfy the error condition $- \langle \varepsilon^k, \bar v^k \rangle \leq \sigma \Vert \bar v^k\Vert^2$.
> - If the approximation error, $\varepsilon^k$, of the proximal update goes to zero then this error condition will trivially be satisfied.
> - Section 6.1 show that the error condition can be satisfied for one step of the inner prox solver (but this puts a restriction on $\rho>-1/2L$).
> - More inner loops leads to a smaller inexactness $\varepsilon^k$, so the error condition becomes easier to satisfy, which in turn puts a looser requirement on $\rho$ (see Section 6.2).
>
> We hope this helps clarify the benefit of multiple inner steps.
>
> > Extending the range of $\rho$
>
> To clarify, Appendix G mentioned in Section 8 is concerned with single-step approximation methods (which are restricted to $\rho > -\gamma/2$).
> For those methods it is interestingly possible to relax the Lipschitz condition (and in effect the condition on the stepsize $\gamma$).
> We are not claiming that we can extend $\rho$ beyond $-1/L$.
>
> > Application to real-world problems
>
> The work is theoretical in nature and follows a long list of previous theoretical works in this space (e.g. [1,2,3,4]), where it is the practice to not include any experience.
> In contrast, we verify our theoretical findings on a toy example.
>
> With that said, Algorithm 1 is designed with practical concerns in mind:
>
> - removing the logarithmic factor is important in practice, where a logarithm could otherwise make a method multiple times slower
> - The method automatically adapts when favorable structure is present (Section 7)
> - the method automatically selects the number of inner steps and the stepsize $\alpha_k$ (which otherwise would need to be tuned)
>
> We hope these points help clarify the practical relevance of our contributions, even within the theoretical focus of this work.
>
> [1] https://openreview.net/pdf?id=lWy2lCTyJa
>
> [2] https://openreview.net/pdf?id=EK7fuAMNoI
>
> [3] https://arxiv.org/pdf/2011.00364
>
> [4] https://arxiv.org/pdf/2210.13831
>
>
> > Unified notation
>
> We are using the following convention:
>
> - When presenting algorithms (and the associated theorems) we state parameters of the algorithm as a function of the problem parameters (e.g. specify $\gamma$ based on $\rho$)
> - When discussing the range of $\rho$ we explicitly state as function of problem parameters (e.g. the introduction and remarks such as Remark 4.1, Remark 5.2).
>
> For consistency we now also mention the range of $\rho$ in Remark 6.4 (after stating the complexity result). We hope this clarifies the choice of presentation.

---

> > ### Comment · Reviewer_Gq7N · 2024-11-26
> >
> > Thank you for your detailed responses. I have decided to maintain my score.

---

### Official Review · Reviewer_kMEF · 2024-11-04

**Soundness:** 3
**Presentation:** 3
**Contribution:** 2
**Rating:** 6
**Confidence:** 3

**Summary:**

This paper investigates variational inequalities that satisfy the weak Minty variational inequality (MVI) condition and proposes a hybrid proximal extragradient method. The authors establish improved complexity bounds for both the implicit and computationally efficient explicit versions of this method, extending previous results to a broader range of the MVI parameter $\rho$. The tightness of this range is confirmed through a lower bound analysis for the exact version of the method. Additionally, linear convergence is shown to be achievable under appropriate conditions.

**Strengths:**

1. **Theoretical Results**: This paper achieves superior complexity and convergence over a broader range of $\rho$. In particular, the authors establish an upper bound for the number of inner iterations that is independent of target accuracy. The theoretical results are further enriched by a lower bound analysis, thoroughly addressing the questions posed in the introduction.
2. **Algorithm Design**: The authors construct the algorithm based on a halfspace projection operator, yielding a hybrid method that combines the inexact proximal point method with the extragradient method. This novel perspective offers a systematic approach to determining the step size and recovers several algorithms from prior work under specific conditions.
3. **Presentation**: The paper is well-organized, easy to follow, and contains extensive content.

**Weaknesses:**

1. **Significance (major concern)**: While this paper addresses the questions raised in line 47, the practical significance may be limited. Previous work has achieved $O(1 / \epsilon \log (1 / \epsilon) )$ complexity for $\rho > -1/L$; the marginal gain from removing the logarithmic term (which is much smaller than $1/\epsilon$) is unclear. Additionally, the $O(1/\epsilon)$ complexity is already achievable for $\rho > -1/(2L)$, so the practical value of studying $\rho \in (-1/L, -1/(2L)]$—a narrow range—seems limited.

2. **Presentation Complexity**: The relationship between the proposed algorithm and existing methods is somewhat complex. A diagram could help clarify this relationship. Additionally, the placement of Algorithm 1 feels distant from Section 6.

**Questions:**

1. The authors refer to the halfspace projection as an *error correction* step. When comparing Eqs. 9 and 10, if we substitute the first equality condition from Eq. 8 into Eq. 9, we get $z^{k+1} = (1-\lambda_k) z^k + \lambda_k (z^k - \bar{v}^k - \epsilon^k)$. Can we interpret the error correction as eliminating the *explicit* error term $\epsilon^k$ in Eq. 10 through an additional operator evaluation $\bar{v}^k \in \gamma S \bar{z}^k$? Consequently, although the calculation of $\bar{z}^k$ involves this error term, it only appears *implicitly*, and with the error condition, it ensures Fejér monotonicity.

2. Why is the parameter $\lambda_{k}$ considered within $(0,2)$? The analysis suggests that $\lambda_{k}$ primarily affects the step size $\lambda_k \alpha_k$, while the theoretical results indicate that $\lambda_k=1$ yields the best convergence. What are the benefits of considering $\lambda_k < 1$ or $\lambda_k > 1$?

3. Does $\alpha_k$ have an upper bound?

---

> ### Author Response · Authors · 2024-11-20
>
> We thank the reviewer for the feedback and address all remaining concerns below.
>
> > Significance
>
> The paper solves an open problem stated by at least two different groups (see e.g. [1] and [2]).
> Furthermore, the solution seems to be nonobvious, as there is a range of work either not getting the full range of $\rho$ [Fan et al. 2023] or suffering the additional log factor [Alacaoglu et al. 2024, Lee and Kim 2024].
> The work addresses a fundamental question about the limits of first-order methods in nonmonotone problems by obtaining what has been theorized to be the largest range of $\rho$ and best complexities for weak MVI.
> It thus seems interesting to the optimization for machine learning community.
>
> In addition:
>
> - Our work provides a unification of many existing methods (we have now included an explicit list in Appendix B)
> - We provide linear convergence (i.e. exponential convergence) under the error bound condition. This is e.g. satisfied by the weak MVI lower bound example in [Pethick et al. (2022)].
> - Algorithm 1 does not need to prespecify the number of inner steps $n$ or the stepsize $\alpha_k$ in contrast with relaxed inexact proximal based methods (see Remark 6.4)
>
> > Clarify the relationship with existing methods
>
> We have added an appendix (Appendix B) clarifying how to obtain existing algorithms such as the relaxed proximal point, the method of Solodov & Svaiter, AdaptiveEG+, forward-backward-forward, relaxed extragradient/extragradient+ from our unified framework.
> Each of the methods are presented in the notation of the paper to make the comparison clear.
>
> > "The placement of Algorithm 1 feels distant from Section 6."
>
> We have moved the algorithm to the section titled "Algorithmic construction" (Section 4).
>
> **Questions**:
>
> > 1. "The authors refer to the halfspace projection as an *error correction* step. When comparing Eqs. 9 and 10, if we substitute the first equality condition from Eq. 8 into Eq. 9, we get $z^{k+1} = (1-\lambda_k) z^k + \lambda_k (z^k - \bar{v}^k - \epsilon^k)$. Can we interpret the error correction as eliminating the *explicit* error term $\epsilon^k$ in Eq. 10 through an additional operator evaluation $\bar{v}^k \in \gamma S \bar{z}^k$? Consequently, although the calculation of $\bar{z}^k$ involves this error term, it only appears *implicitly*, and with the error condition, it ensures Fejér monotonicity."
>
> Yes, the fact that $\varepsilon^k$ does not explicitly appear in the update of $z^{k+1}$ is one way of intuitively understanding why we can eventually ensure Fejér monotonicity.
> Conversely, it also shows why relaxed inexact proximal point suffers the logarithmic factor in the complexity.
>
> > 2. "Why is the parameter $\lambda_{k}$ considered within $(0,2)$?"
>
> Instead of only considering $\lambda_k=1$ we consider the wider range $(0,2)$ for two reasons:
>
> - Mainly, it allows us to absorb adaptive $\alpha_k$ into $\lambda_k$, which is necessary for capturing the constant stepsize case (see Section 6.3).
> - Overrelaxation (i.e. $\lambda_k \in (1,2)$) has been shown to be beneficial in practice (see e.g. [3]), so it is useful to have guarantees for the extended range.
>
> [3] https://arxiv.org/pdf/1603.05398
>
> > 3. Does $\alpha_k$ have an upper bound?
>
> Yes, by rewriting
>
> $\alpha_k = 1 + \langle\bar v, \varepsilon\rangle/\Vert\bar v\Vert^2 + \delta/\gamma \leq 1 + \Vert\bar v\Vert \Vert \varepsilon\Vert/\Vert\bar v\Vert^2 + \delta/\gamma = 1 + \delta/\gamma$
>
> where we have used Cauchy-Schwarz and minimized with $\varepsilon=0$.

---

> > ### Comment · Reviewer_kMEF · 2024-11-26
> >
> > I thank the authors for their detailed responses. I decide to maintain my score.

---

### Meta-Review · Area_Chair_BrGd · 2024-12-22

**Metareview:**

The paper studies nonmonotone games that satisfy the weak Minty variational inequality assumption with parameter $\rho$. The paper proposes a hybrid proximal extragradient algorithm for the problem with an improved convergence rate of $O(1/eps)$ in a broader range of $\rho$ values, which is an improvement over prior works by a $\log(1/\epsilon)$ factor. The paper also shows that the algorithm achieves a linear convergence rate under some assumptions.

The main contribution is a valuable addition to the area of nonmonotone games, and it strengthens and unifies the prior convergence results. Additionally, removing the extra logarithmic factor from the convergence rate seems challenging. The reviewers were concerned that the logarithmic factor improvement is only in a narrow range of $\rho$ parameter values, which makes the result limited in scope. Additionally, the reviewers were concerned that the contribution is primarily theoretical and the potential practical applications seem unclear. Although the contribution may be somewhat limited, this work makes a valuable theoretical contribution to this line of work and closes a gap in the literature.

**Additional Comments On Reviewer Discussion:**

The reviewers asked several clarifying questions that were addressed by the authors. There was also a discussion on possible extensions to other settings, such as the stochastic setting. These extensions are not considered in the current work.

---

### Decision · Program_Chairs · 2025-01-22

Accept (Poster)